# Fleet Supervisor Allocation: A Submodular Maximization Approach

**Oguzhan Akcin, Ahmet Ege Tanriverdi, Kaan Kale and Sandeep P. Chinchali**
The University of Texas at Austin
{oguzhanakcin, aet2776, kaankale, sandeepc}@utexas.edu

**Abstract:** In real-world scenarios, the data collected by robots in diverse and unpredictable environments is crucial for enhancing their perception and decision-making models. This data is predominantly collected under human supervision, particularly through imitation learning (IL), where robots learn complex tasks by observing human supervisors. However, the deployment of multiple robots and supervisors to accelerate the learning process often leads to data redundancy and inefficiencies, especially as the scale of robot fleets increases. Moreover, the reliance on teleoperation for supervision introduces additional challenges due to potential network connectivity issues. To address these issues in data collection, we introduce an Adaptive Submodular Allocation policy, ASA, designed for efficient human supervision allocation within multi-robot systems under uncertain connectivity conditions. Our approach reduces data redundancy by balancing the informativeness and diversity of data collection, and is capable of accommodating connectivity variances. We evaluate the effectiveness of ASA in simulations with 100 robots across four different environments and various network settings, including a real-world teleoperation scenario over a 5G network. We train and test our policy, ASA, and state-of-the-art policies utilizing NVIDIA's Isaac Gym. Our results show that ASA enhances the return on human effort by up to $3.37\times$, outperforming current baselines in all simulated scenarios and providing robustness against connectivity disruptions.

**Keywords:** Imitation Learning, Submodular Maximization, Fleet Learning

## 1 Introduction

Today, diverse industries deploy robotic fleets for tasks ranging from autonomous driving [1, 2] to healthcare [3] and package delivery [4]. These robots are often deployed with policies trained on a dataset that is primarily based on simulations, along with a small amount of data collected through real-world interactions. While effective within their training contexts, these models often fail to adapt to new or evolving real-world scenarios [5], making data collection critical for the success of the robotics applications [6, 7].

A popular approach to collecting such data is through human supervision, where humans directly guide the robots to perform the tasks. These data are then used to train the robots via Imitation Learning (IL), where the robots learn to perform tasks by imitating the human demonstrations [8]. Imitation Learning (IL) has been effective in many robotics applications, ranging from autonomous driving [9] to robotic manipulation [10, 11]. However, the breadth of scenarios necessary for effective IL emphasizes the need for continual data collection [12], commonly done with numerous robots in parallel. Usually, the number of humans is less than that of deployed robots. For instance, a recent autonomous delivery company, Starship Technologies, operates 1700 autonomous robots while teleoperating only $1\%$ of this robotic fleet [13, 14]. The scarcity of human supervisors necessitates the selection of informative robots for supervision [15, 16].

Human supervision is often provided through real-time teleoperation over a network, especially when supervising fleets of robots distributed across the globe. For example, various companies, including Cruise, utilize human supervisors located in their control centers to teleoperate autonomous

8th Conference on Robot Learning (CoRL 2024), Munich, Germany.

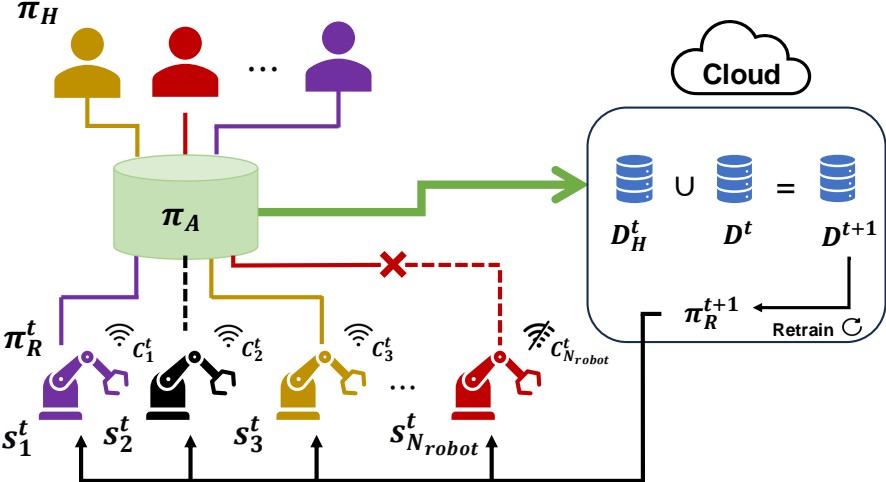

Figure 1: **Supervisor Allocation Problem.** At each time step $t$, the human supervisors with policy $\pi_H$ are allocated to the robots with policy $\pi_R^t$ based on the allocation policy $\pi_A$. Each robot $i$ operates in a different state $s_i^t$, and the human supervisors are allocated to the robots based on the uncertainty of the robots and the similarity between the robots. Additionally, the connections to the robots may not be stable, which is indicated by the connection random variable, $C_i^t$, of each robot $i$. At the end of each time step $t$, the data collected by human supervisors $\mathbf{D}_H^t$ is added to the dataset $\mathbf{D}^t$ to create an updated dataset $\mathbf{D}^{t+1}$, which is then used to train the robot policy $\pi_R^{t+1}$.

vehicles deployed across the world [17, 18]. However, these networks might be susceptible to connection failures [19], and it is important to be robust against network uncertainties. Combining these challenges with selecting informative robots, we formulate the Supervisor Allocation Problem ( Fig. 1), which involves managing the limited human resources to maximize data diversity and quality under uncertain network connectivity. Our problem is an extension of the Interactive Fleet Learning (IFL) setting introduced by [15]. We extend the IFL setting to account for network elements that play an important role in real-world teleoperation scenarios [20].

We then introduce a novel human supervisor allocation policy called Adaptive Submodular Allocation (ASA). ASA distributes human supervisory capacity across a fleet, ensuring a balance of data informativeness and diversity to minimize redundancy in data collection. Our allocation policy is shown to be robust against network instabilities and is able to adapt to the dynamic nature of data collection, which we demonstrate through extensive simulations in diverse network environments, including real-world 5G scenarios. We show that ASA improves the Return on Human Effort (RoHE) [15], which is a human supervision efficacy metric, by up to $3.37\times$ compared to existing benchmarks.

## 2  Related Work

Data collection is a critical problem in robotics and machine learning that is essential for continually improving the performance of robots [21–25]. It is closely related to active learning [26–30], where the goal is to select the most informative samples to label. Although the goal of data collection is similar to active learning, the focus is on collecting data samples that are the most informative for training the models. In our case, however, the aim is to select the robots that provide the most informative data for human supervisors.

IL is a popular approach in robotic learning, where robots learn policies from human demonstrations [31–34]. Despite its potential, the reliance on purely offline data introduces several challenges, such as distribution shifts [35], which occur when robots encounter states that were not previously experienced by humans. These issues can be alleviated through online data collection methods, such as Dataset Aggregation (DAgger) [35] and various forms of interactive IL [36, 37]. Most interactive IL methods rely on human supervision to decide when to intervene in the robot's learning process. This presents scalability challenges, especially when applied to extensive robot networks [38] or during prolonged learning phases [39]. Robot-initiated interactive IL strategies like EnsembleDAgger [40] and ThriftyDAgger [41] have been proposed to mitigate these constraints, enabling robots to request

human input under specific conditions. However, these methods are designed for single-robot task allocation scenarios and do not consider multi-robot scenarios. Closest to our work, Fleet-DAgger [15] has been proposed to address the supervisor allocation problem in a multi-robot scenario. However, Fleet-DAgger does not consider operational constraints that might limit the allocation of human supervisors, such as network connectivity and the potential redundancy from employing multiple human supervisors in similar environments. Our work, on the other hand, focuses on learning an allocation policy that is adaptable to the operational constraints while minimizing the redundancy in the data collection, which is crucial for the system's scalability [27].

One popular approach to mitigate redundancy in data collection is using submodular maximization. Submodularity refers to the property of the marginal gain of adding an item to a small set being higher than adding the same item to a large set. As submodularity is a common trend in data collection, it has been widely used in machine learning tasks, such as sensor placement [42], active learning [27, 30, 43], and summarization [44]. Submodular maximization has also been extended to stochastic settings [45, 46], where the goal is to select a subset of items to maximize the expected value of a submodular function. Despite its wide use in machine learning, stochastic submodular maximization has not been used in the context of IL and multi-robot data collection scenarios. Our work is the first to use stochastic submodular maximization in the context of human supervision and multi-robot scenarios to address the supervisor allocation problem.

## 3 Problem Formulation

Consider a geo-distributed system of $N_{\text{robot}}$ robots, $\mathbf{I} = \{1, \cdots, N_{\text{robot}}\}$. Each robot $i$ operates in parallel within an independent Markov Decision Process (MDP) with a different initial state. However, all robots operate within the same state and action spaces $\mathbf{S}$ and $\mathbf{A}$, respectively. Each robot $i$ observes the state of the environment $s_i^t \in \mathbf{S}$ at time $t$ and selects an action $a_i^t \in \mathbf{A}$ based on a policy $\pi_{\text{R}}^t : \mathbf{S} \to \mathbf{A}$. The robots share the same policy $\pi_{\text{R}}^t$ that has been trained using the collective data $\mathbf{D}^t$ accumulated up to time step $t$. We define the collection of states and actions for all robots as $\mathbf{s}^t = (s_1^t, \ldots, s_{N_{\text{robot}}}^t) \in \mathbf{S}^{N_{\text{robot}}}$ and $\mathbf{a}^t = (a_1^t, \ldots, a_{N_{\text{robot}}}^t) \in \mathbf{A}^{N_{\text{robot}}}$. These robots can be supervised by $N_{\text{human}}$ human supervisors with an oracle policy $\pi_{\text{H}} : \mathbf{S} \to \mathbf{A}_H$, respectively. In addition to the robot action space $\mathbf{A}$, the human action space $\mathbf{A}_H$ includes a reset action, which can return the robot to a safe state. Only robots supervised by humans operate with human policy $\pi_{\text{H}}$ while the rest of the robots operate with the robot policy $\pi_{\text{R}}^t$.

**Supervisor Allocation and Connectivity:** In each time step $t$, $N_{\text{human}}$ human supervisors can be assigned to the robots for assistance. However, the connections to the robots are unreliable, with $C^t = \{C_1^t, \cdots, C_{N_{\text{robot}}}^t\} \in \{0,1\}^{N_{\text{robot}}}$ denoting independent random variables associated with the connection reliability of the robots. $C_i^t \in \{0,1\}$ indicates whether a successful connection with robot $i$ can be established ($C_i^t = 1$) or not ($C_i^t = 0$) at time $t$. Under this uncertain connectivity, we are interested in finding an allocation policy $\pi_{\text{A}} : \{0,1\}^{N_{\text{robot}}} \times \mathbf{S}^{N_{\text{robot}}} \times \mathbf{A}^{N_{\text{robot}}} \times \mathbf{I} \to 2^{\mathbf{I}}$ that selects robots to be supervised $X \subseteq \mathbf{I}$ based on connection reliability $C^t$, collection of states $\mathbf{s}^t$ and actions $\mathbf{a}^t$. Our setting extends the Interactive Fleet Learning setting [15] by incorporating imperfect time-varying network connectivity.

**Data Collection and Policy Retraining:** Upon allocation, human supervisors contribute data only from successful connections, forming the human supervision data $\mathbf{D}_H^t$. This new data is integrated into the current dataset and the robot policy $\pi_{\text{R}}^t$ is retrained using a retraining function $g$:

$$\mathbf{D}^{t+1} = \mathbf{D}^t \cup \mathbf{D}_H^t, \quad \mathbf{D}_H^t = \{(s_i^t, \pi_{\text{H}}(s_i^t)) : i \in X \text{ and } C_i^t = 1\}, \tag{1}$$

$$\pi_{\text{R}}^{t+1} = g(\pi_{\text{R}}^t, \mathbf{D}^{t+1}). \tag{2}$$

**Objective:** Our objective is to develop an allocation policy $\pi_{\text{A}}$ that maximizes the expected Return on Human Effort (RoHE) over the connectivity $C^t$. RoHE metric was introduced along with Interactive Fleet Learning setup [15] to set a benchmark in Fleet Learning settings. It is a ratio of the total reward obtained by the fleet to the total number of human actions. Formally, the objective is to maximize the expected RoHE over the connection probabilities:

$$\max_{\pi_{\text{A}} \in \Omega} \mathbb{E}_C \left[ \frac{N_{\text{human}}}{N_{\text{robot}}} \frac{\sum_{i \in \mathbf{I}} \sum_{t=0}^{T} r(s_i^t, a_i^t)}{1 + \sum_{t=0}^{T} |\pi_{\text{A}}(C^t, \mathbf{s}^t, \mathbf{a}^t, \mathbf{I})|_F^2} \right]. \tag{3}$$

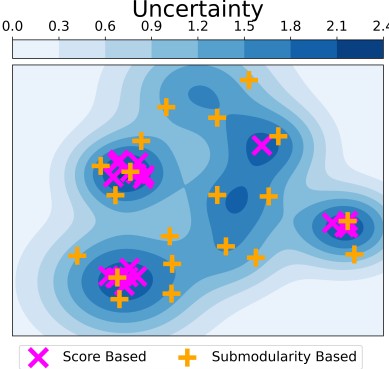

Figure 2: **Submodular maximization balances uncertainty and diversity.** This figure illustrates a toy example of our allocation problem in a 2D state space. The blue contours indicate the uncertainty levels, with darker shades representing higher uncertainty. Purple crosses (traditional score-based allocations) and yellow plus signs (our submodularity-based allocations) mark the positions of selected robots. Unlike score-based methods that often select highly uncertain but potentially overlapping states, our approach strategically picks a more diverse set of states, effectively balancing the trade-off between high uncertainty and coverage, thereby reducing data redundancy and enhancing the training data's representativeness.

Here, $C = \{C^0, \ldots, C^T\}$ represents the set of all connection random variables from time 0 to time horizon $T$. The function $r : \mathbf{S} \times \mathbf{A} \to \mathbb{R}$ is the reward function, $|\cdot|_F$ denotes the Frobenius norm, and $\Omega$ refers to a set of all allocation policies. Intuitively, RoHE measures the overall performance of the robotic fleet, normalized by the total number of human interventions.

## 4   A Stochastic Submodular Maximization Approach

We now present our novel policy, adaptive submodular allocation (ASA), for the problem outlined in Eq. 3. First, we define the stochastic submodular maximization problem, which represents the value of robot supervision, and then we define the greedy algorithm, which is used to pick the robots to supervise.

### 4.1   Submodular Maximization Problem

To address the optimization problem presented in Eq. 3, we use stochastic submodular maximization. Stochastic submodular maximization is particularly suited to our scenario because it leverages the diminishing returns property that naturally reflects the decrease in the marginal gain of supervising additional robots. Furthermore, the method inherently discourages the selection of similar robots, thereby avoiding the assignment of humans to robots that offer overlapping information, which decreases the return on human effort. Finally, stochastic submodular maximization accounts for the inherent network uncertainty in our problem by acknowledging the non-deterministic connections to the robots. This is crucial for developing a robust solution across different connectivity patterns.

We first define a submodular objective function $f_{C^t} : 2^{N_{\text{robot}}} \to \mathbb{R}$ that quantifies the value of supervising a selected set of robots $X$, considering the allocation reliability outcomes for these robots $C^t$. We define our objective function based on the facility location problem, a classic example of a submodular maximization objective [42], as follows:

$$f_{C^t}(X) = \sum_{i \in \mathbf{I}} \max_{j \in X} C_j^t M_{j,i}^t. \tag{4}$$

Here, $X$ is the set of robots selected for human supervision, and $C_j^t$ indicates whether the connection to the robot $j$ is successful or not. $M_{j,i}^t$ represents the value of supervision of the robot $j$ on the robot $i$ at time $t$, and we consider two factors: the informativeness of the robot $i$ and the similarity between the robot $j$ and the robot $i$. Additionally, our formulation is modular and can be extended to include other factors, such as prioritizing the robots that have violated the safety constraints or those in critical states. With all factors combined, we define the value of supervision $M_{j,i}^t$ as:

$$M_{j,i}^t = \mathcal{S}^t(j,i) * \mathcal{U}^t(i) + \mathcal{K}^t(i). \tag{5}$$

Here, $\mathcal{S}^t(j,i)$ defines the similarity between the robots $i$ and $j$ at time $t$, and $\mathcal{U}^t(i)$ is the informativeness of the robot $i$, while $\mathcal{K}^t(i)$ is an indicator of whether the robot $i$ violates the safety constraints or is in a critical state. Our definition of $M_{j,i}^t$ is modular, and each factor can be defined based on specific requirements. For example, the similarity function $\mathcal{S}^t$ can be defined as the cosine similarity, the Euclidean distance, or any other similarity metric. The informativeness of the robot $\mathcal{U}^t(i)$

can be defined as the entropy of the robot's policy or the uncertainty of the robot's state, while the constraint function $\mathcal{K}^t(i)$ can be defined based on the safety constraints or the critical states for the robots, similar to previous works [15, 47]. For the exact definitions of similarity, informativeness, and constraint functions, please refer to the Appendix. With the objective function defined, we pose the following maximization problem to optimize our allocation policy:

$$\max_{X \subseteq \mathbf{I}} \quad F(X), \tag{6}$$

$$\text{subject to: } |X| \leq N_{\text{human}},$$

where the goal is to identify the subset of robots $X$ that maximizes the expected value of $f_{C^t}$, denoted as $F(X) = \mathbb{E}_{C^t}[f_{C^t}(X)]$ over all possible connection outcomes $C^t$. The constraint $|X| \leq N_{\text{human}}$ ensures that the number of selected robots is limited by the number of available human supervisors $N_{\text{human}}$.

### 4.2 Adaptive Submodular Allocation (ASA) Policy

Now, we can present our allocation policy ASA based on a greedy algorithm given in Algorithm 1. Starting from an empty solution set $X$ (line 1), ASA iteratively selects the robot with the highest marginal gain based on the estimated expected value function $\hat{F}$ (line 3), which estimates the expected value of the submodular objective function $f$. Then, ASA computes the expected marginal gain of selecting the robot $x^*$ (line 4), and if the expected marginal gain is below a certain threshold, the algorithm stops the selection process (line 5). This threshold ensures that the algorithm avoids using unnecessary human effort by stopping when the marginal gain of selecting an additional robot is low. Otherwise, the chosen robot $x^*$ is added to the solution set $X$ (line 8). Finally, if there is an observation on whether the connection to the robot was successful or not, the estimated expected value function $\hat{F}$ is updated (line 9).

The estimated expected value function $\hat{F}$ is an important factor in our allocation policy and can be defined as:

$$\hat{F}(X) = \sum_{\xi} f_{\xi}(X)\hat{p}_{\xi} = \sum_{\xi} f_{\xi}(X) \prod_{i \in \mathbf{I}} \hat{p}_{\xi_i}. \tag{7}$$

Here, the function $f_{\xi}$ denotes the value of supervising the set of robots $X$ for a specific realization $\xi$ of the connection random variable $C^t$. $\hat{p}_{\xi}$ represents the estimated connection probability of the realization $\xi$, whereas $\hat{p}_{\xi_i}$ denotes the estimated connection probability for the robot $i$ within the realization $\xi$. Initially, $\hat{p}_{\xi_i}$ values are set to the actual initial connection probabilities $\hat{p}_{\xi_i} \leftarrow P(\xi_i = C_i^0)$ for all robots $i$. Then, if an observation on the success of supervisor allocation is made, the connection probability estimate $\hat{p}_{\xi_i}$ is updated for the selected robot $i$: $\hat{p}_{\xi_i} \leftarrow P(\xi_i = C_i^t)$.

Another important factor in the allocation policy is the marginal threshold parameter, which determines the trade-off between the cost of additional supervision and the incremental benefit derived from including an additional robot in the supervision set. In practice, this threshold can be set to 0 if all human supervisors want to be allocated to maximize the amount of data collected. Then, the threshold can be gradually increased to optimize the efficiency of human supervision.

Based on the availability of the observations of the connection probabilities, we define two variants of our policy: non-Adaptive Submodular Allocation (n-ASA) and Adaptive Submodular Allocation (ASA). In n-ASA, we are not able to observe the connection probabilities, and thus, the allocations are done beforehand. In ASA, on the other hand, the robots are selected iteratively based on the success of the allocations, and the connection probability estimates $\hat{p}_{\xi}$ are

---

**Algorithm 1** ASA Policy

**Input:** Estimated expected value of submodular objective function $\hat{F}$, set of all robots $\mathbf{I}$
**Output:** robots selected for supervision $X$

1: Initialize $X \leftarrow \emptyset$
2: **for** $k = 1$ to $N_{\text{human}}$ **do**
3:     $x^* \leftarrow \text{argmax}_{x \in \mathbf{I} \setminus X} \hat{F}(X \cup \{x\})$
4:     Compute $\Delta \leftarrow \hat{F}(X \cup \{x^*\}) - \hat{F}(X))$
5:     **if** $\Delta <$ threshold **then**
6:         **break**
7:     **end if**
8:     $X \leftarrow X \cup x^*$
9:     **if** ASA **then**
10:         Observe whether connection to robot $x^*$ is successful and update $\hat{F}$
11:     **end if**
12: **end for**

updated. To visualize the differences between n-ASA and ASA, consider the following: in both cases, robot 1 is selected for supervision in the first iteration. While selecting the robot to supervise, n-ASA considers both possibilities (successful and unsuccessful connection to robot 1) and selects the second robot that maximizes the expected marginal gain over both cases. However, in ASA, after selecting robot 1 for supervision, we observe whether the connection was successful or not and select the next robot that maximizes the expected marginal gain based on the observation. We then update the connection probability estimate of the robot 1 with its actual probability: $\hat{p}_{\xi_1} \leftarrow P(\xi_1 = C_1^t)$.

When the marginal threshold parameter is set to zero and the estimated expected value function $\hat{F}$ is equal to the actual expected value of supervision $F$, ASA is equivalent to the greedy algorithm for submodular maximization, which is proven to approximate the optimal solution for the submodular maximization problem in Eq. 6 with a factor of $1 - 1/e$ [45]. Additionally, n-ASA approximates the optimal adaptive policy with a factor of $(1 - 1/e)^2$ [45]. To compute selected robots faster, we use the lazy greedy algorithm [46]; this has the same time complexity as Algorithm 1 but has a better empirical performance.

## 5   Experiments

We consider a fleet of $N_{\text{robot}} = 100$ robots that can be supervised by $N_{\text{human}} = 5$ human supervisors. The human supervisors are implemented as reinforcement learning agents using the Proximal Policy Optimization (PPO) algorithm [48]. We utilize the behavior cloning algorithm to initialize the robot policies based on an offline dataset of 5000 state-action pairs and use our allocation policy to collect data from the robots and update the models. In all of our experiments, when the robots violate the constraints, we perform a hard reset to bring the robots back to a safe state. We set the hard reset time to $t_R = 5$ timesteps, the minimum intervention time to $t_T = 5$ timesteps, and the fleet operation time to $T = 10,000$ timesteps. We average the results over 3 random seeds for each task and network configuration. We have chosen these parameters to align with the settings used in the environments of the benchmark algorithms [15] for a fair comparison.

**Environments:** We consider four different environments in our experiments. First is Humanoid, where the robots focus on bipedal locomotion. Second is ANYmal, which involves quadruped locomotion using the ANYmal robot. Third is Allegro Hand, focusing on dexterous manipulation tasks. The last environment is Ball Balance and it requires the robots to balance a ball on a plate.

**Network Configurations:** We use five different network configurations based on various connectivity probabilities. **Always** allows robots to always be supervised by human supervisors. In **Mixed-Scarce**, some robots have a high connection probability, while others have a low probability. **Ookla** features varying connection probabilities according to cellular network performance metrics [49]. **5G** uses real-world connectivity data, including latency and throughput, collected from a university robotics lab (further details in the Appendix). Finally, **Changing-Scarce** starts with the same connection probabilities as Mixed-Scarce but evolves over time. It is important to note that while the first four configurations maintain stationary network connectivity, the Changing-Scarce configuration introduces dynamic network connectivity that changes over time.

**Metrics:** We evaluate the performance of the allocation policies based on the following metrics: (1) Return on Human Effort (RoHE), which is given in Eq. 3 and (2) the cumulative number of successfully completed tasks by the entire fleet, which we will refer to as cumulative success. To simplify, RoHE measures the fleet performance per human intervention, while cumulative success only considers the total successful task completion without considering the number of interventions. For example, simply allocating all human supervisors would improve cumulative success but decrease RoHE. An ideal allocation policy should balance the two, as an ideal system would require a high total success while using humans as efficiently as possible.

**Allocation Policies:** We compare our proposed **Adaptive Submodular Allocation (ASA)** and **non-Adaptive Submodular Allocation (n-ASA)** policies with several baselines. The ASA policy uses the observed connection information to update the network connection probabilities, while the n-ASA policy does not use this information. The **Random** baseline simply selects robots for supervision randomly. **Fleet-EnsembleDAgger (FE)** uses variance for uncertainty estimation, combined with constraint-based prioritization [40]. **Fleet-ThriftyDAgger (FT)** integrates uncertainty and goal-oriented prioritization, adapting ThriftyDAgger for fleet settings in goal-oriented environments [41]. **Fleet-DAgger (FD)** prioritizes robots violating constraints and selects those with the

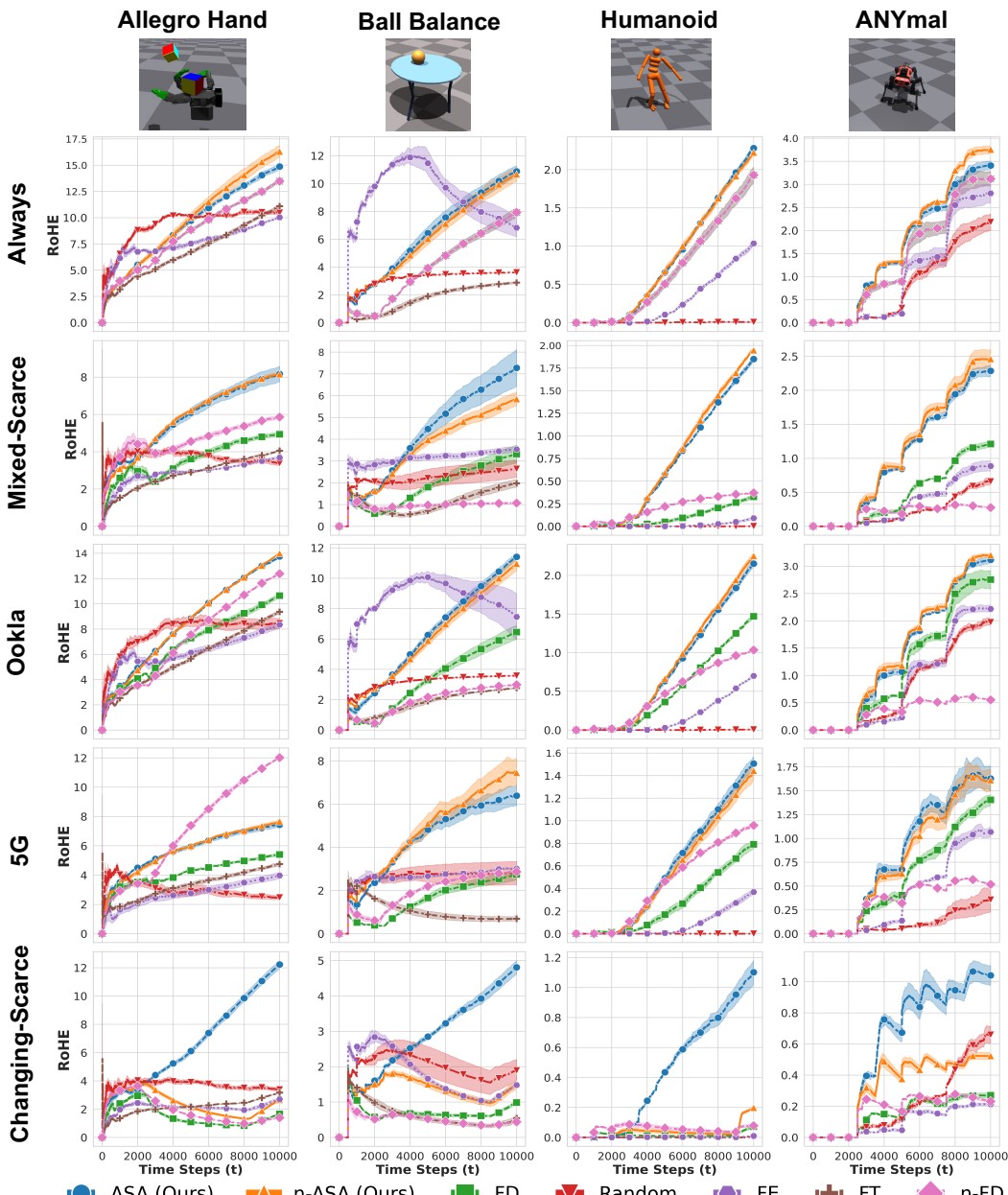

Figure 3: **Our ASA and n-ASA policies outperform other benchmarks across all environments and network combinations.** Here, each row represents a different network configuration, and each column corresponds to a different environment. Compared to other policies, ASA and n-ASA are affected less by changes in the network configurations because of their stochastic submodular maximization-based policies that can incorporate network uncertainties. The submodular maximization objective improves the performance when there are no network uncertainties (row 1) due to its ability to cover diverse and informative scenarios. Additionally, ASA outperforms n-ASA in the Changing-Scarce network configuration (row 5) due to its adaptive nature to network connectivity changes.

highest uncertainty and risk of failure [15]. Lastly, **n-Fleet-DAgger (n-FD)** adapts Fleet-DAgger to network uncertainties by filtering out robots with low connection probabilities.

**How do ASA and n-ASA perform under different network configurations?**

We evaluate the performance of our policies, ASA and n-ASA, under different network configurations for each environment. The RoHE metric for each time step has been shown in Fig. 3, and cumulative success values at the final time step have been presented in Table 1. In both metrics, we can see that our ASA and n-ASA policies are able to outperform other benchmarks. This is

| Allocation Policy | Allegro Hand | | AnyMAL | | Ball Balance | | Humanoid | |
|---|---|---|---|---|---|---|---|---|
| | RoHE | Cumulative Success | RoHE | Cumulative Success | RoHE | Cumulative Success | RoHE | Cumulative Success |
| Random | 5.34 | 387.94 | 1.04 | 92.66 | 2.90 | 1448.73 | 0 | 0 |
| FT | 6.49 | 2119.53 | - | - | 1.77 | 956.53 | - | - |
| FE | 5.80 | 1774.47 | 1.44 | 115.47 | 4.46 | 1153.20 | 0.44 | 150.27 |
| FD | 7.23 | 2251.06 | 1.77 | 171.60 | 4.28 | 1175.2 | 0.76 | 266.80 |
| n-FD | 9.02 | 3723.80 | 1.07 | 195.40 | 3.05 | 1035.53 | 0.87 | 329.07 |
| n-ASA (Ours) | 9.75 | 3729.50 | **2.31** | 232.47 | 5.10 | 1167.27 | 1.61 | 408.07 |
| ASA (Ours) | **11.29** | **4217.70** | 2.30 | **234.27** | **8.15** | **1636.53** | **2.93** | **478.6** |

Table 1: **Average RoHE and cumulative success of different allocation policies.** As shown by the RoHE and cumulative success values averaged over all network configurations, our ASA and n-ASA policies outperform other baselines in all environments (columns). This shows the adaptability of our methods to different environments and network configurations in terms of efficient use of human effort and high task completion.

because our policies can incorporate network uncertainty information into their allocation algorithm through stochastic submodularity. On the other hand, other benchmark policies, except n-FD, are not designed to incorporate such network uncertainty. The n-FD policy is able to incorporate network uncertainties and outperforms our policies in the 5G network for the Allegro Hand task, but it fails to generalize to other network configurations and tasks. We can see that our allocation policy outperforms other baselines in terms of the RoHE metric by up to $1.25\times$, $1.31\times$, $1.83\times$, and $3.37\times$ in Allegro Hand, ANYmal, Ball Balance, and Humanoid environments, respectively.

**How do ASA and n-ASA compare when the network connectivity is stable?**

To test whether our RoHE gains are only due to adaptability to different network configurations, we have also simulated a network where all robots are always reachable. In row 1 of Fig. 3, we can see that our ASA and n-ASA policies still outperform other allocation benchmarks due to their ability to diversify the selected robots to cover more states. As we have shown in the 2D toy example in Fig. 2, rather than only focusing on the states with high uncertainty, ASA and n-ASA consider the whole state space to collect combined data that is more informative.

**How do ASA and n-ASA compare to benchmarks with time-varying connection probabilities?**

To test the adaptability of our allocation policies to network connectivities varying over time, we have run experiments on the Changing-Scarce network scenario. We can see in row 5 of Fig. 3 that the ASA policy consistently outperforms n-ASA, n-FD, and FD, due to its ability to dynamically update the network connection estimates based on the observations on whether the connection to each robot is successful or not. This adaptability is particularly important in scenarios where network conditions are unpredictable and can change over time, such as mobile robotics applications.

**Limitations:** Our work has several limitations. First, it uses only real-world data collected from 5G network without hardware robotics experiments and is tested only in NVIDIA's Isaac Gym environment like previous benchmarks. Additionally, it assumes that network connectivity and robot states and policies are independent across robots, while in real-world scenarios, robots might share the same network or physical location, meaning their policies might affect each other.

# 6 Conclusion and Future Work

We present two novel supervisor allocation policies, ASA and n-ASA, for assigning human supervisors to a robotic fleet for data collection. These policies, based on stochastic submodular maximization, offer a modular approach for incorporating diverse allocation objectives, informativeness metrics, and retraining methods. Our allocation policies beat current benchmarks in RoHE and cumulative success metrics across most environments and network configurations. These performance gains are thanks to their stochastic submodular maximization objective, which incorporates network connectivity while balancing the diversity and informativeness of selected robots. Finally, we collect real-world 5G network data from a field dedicated to teleoperated robots and show the applicability of our allocation policy in real-world scenarios as well.

In a future project, we plan to extend our work to include hardware robotics experiments and other simulation environments, including teleoperation over a 5G network, which would be possible in an application such as autonomous driving. We also plan to investigate the impact of different imitation learning methods to test the generalizability of our allocation policies.

## Acknowledgements

This work was supported by the National Science Foundation under grant no. 2148186 and 2133481 as well as federal agencies and industry partners as specified in the Resilient & Intelligent NextG Systems (RINGS) program. This article solely reflects the opinions and conclusions of its authors and does not represent the views of any sponsor.

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

# A  Appendix

**Code Availability:** The code and related materials can be found in the following code repository:

https://github.com/UTAustin-SwarmLab/Fleet-Supervisor-Allocation.git

The organization of the appendix is as follows:

1. Subsection A.1 describes the details of the submodular maximization function we have used in our experiments.
2. Subsection A.2 describes the details of the theoretical bounds of our allocation algorithm.
3. Subsection A.3 presents the environment setups and exact parameters we have used for our simulations.
4. Subsection A.4 provides the exact details for the network configurations used in our experiments.
5. Subsection A.5 presents all the simulation results for all tasks and network configurations as well as additional metrics providing an insight into why our methods outperform other benchmarks.
6. Subsection A.6 provides ablation studies over the hyperparameters and simulation parameters.
7. Subsection A.7 presents the complexity analysis of our allocation policies.
8. Subsection A.8 presents our 5G network data collection setup.

## A.1  Submodular Maximization Parameters

This section provides the exact parameters and functions used in the submodular maximization method given in Section 4.

### A.1.1  Informativeness Function

The informativeness function $\mathcal{U}^t$ measures the informativeness of each robot $i$ in the fleet at time $t$. To measure the informativeness of the robots, we use a weighted combination of two functions. The first function measures the uncertainty of the robot policy $\pi_{\mathrm{R}}^t$ in the environment, and the second function measures the risk of the robot $i$ violating the constraints $\mathbf{K}$. Then, the informativeness function can be defined as follows:

$$\mathcal{U}^t(i) = \alpha_{\mathcal{U}}\mathcal{U}_{\mathrm{unc}}^t(i) + (1 - \alpha_{\mathcal{U}})\mathcal{U}_{\mathrm{risk}}^t(i). \tag{8}$$

Here, $\alpha_{\mathcal{U}}$ is a hyperparameter that controls the weight of the uncertainty in the overall informativeness measure, and $\mathcal{U}_{\mathrm{unc}}^t(i)$ and $\mathcal{U}_{\mathrm{risk}}^t(i)$ are the uncertainty and risk functions of the robot $i$ at time $t$, respectively. When the robot is taking discrete actions, the uncertainty function $\mathcal{U}_{\mathrm{unc}}^t(i)$ is defined as the entropy of the robot policy $\pi_{\mathrm{R}}^t$, and when the robot is taking continuous actions, the uncertainty function is defined as the ensemble variance of the robot policy $\pi_{\mathrm{R}}^t$ [40]. The risk function $\mathcal{U}_{\mathrm{risk}}^t(i)$ is defined as the likelihood of the robot $i$ exiting the constraint space $\mathbf{K}$ [15].

For both uncertainty and risk functions, if the value of the function for the robot $i$ is below a certain threshold, we set it to zero. We define this threshold parameter as $\mathcal{U}_{\mathrm{unc}}^{\mathrm{thres}}$ and $\mathcal{U}_{\mathrm{risk}}^{\mathrm{thres}}$ for uncertainty and risk functions, respectively. We present specific $\mathcal{U}_{\mathrm{unc}}^{\mathrm{thres}}$ and $\mathcal{U}_{\mathrm{risk}}^{\mathrm{thres}}$ parameters for the experiments in Table 2.

### A.1.2  Similarity Function

The similarity function $\mathcal{S}^t$ measures the similarity between two robots $i$ and $j$ in the fleet at time $t$. In our experiments, we utilize both the similarity between the states and the similarity between the actions taken by the robots. More formally, the similarity function is defined as follows:

$$\mathcal{S}^t(i, j) = \alpha_{\mathcal{S}}\frac{s_i^t \cdot s_j^t}{\|s_i^t\|\|s_j^t\|} + (1 - \alpha_{\mathcal{S}})\frac{a_i^t \cdot a_j^t}{\|a_i^t\|\|a_j^t\|}, \tag{9}$$

where $s_i^t$ and $s_j^t$ are the states of the robots $i$ and $j$, respectively; $a_i^t$ and $a_j^t$ are the actions taken by the robots $i$ and $j$, respectively, at time $t$. $\alpha_{\mathcal{S}}$ is a hyperparameter that controls the weight of the state similarity in the overall similarity measure.

### A.1.3  Constraint Violation Function

The constraint violation function $\mathcal{K}^t(i)$ measures whether robot $i$ violates the constraints at time $t$. In our experiments, we used the constraint violation as an indicator function that returns $\alpha_{\mathcal{K}}$ if the constraint is violated and 0 otherwise. The $\alpha_{\mathcal{K}}$ is a parameter that controls the relative importance of the constraint violation in the

overall objective function. In our experiments, we set $\alpha_{\mathcal{K}} = 10000$ to prioritize the robots with constraint violations. The constraint violation function is defined as follows:

$$\mathcal{K}^t(i) = \begin{cases} \alpha_{\mathcal{K}} & \text{if } s_i^t \notin \mathbf{K}, \\ 0 & \text{otherwise.} \end{cases} \tag{10}$$

Here, the $\mathbf{K}$ refers to the safe states that the robot can operate without any human intervention. As the constraint function causes the system to prioritize the robots violating the constraints, in initial time steps, we set the $\alpha_{\mathcal{K}} = -10000$ to ensure that the robots violating the constraints are not prioritized. This is because, in the initial steps, the robots explore the environment, and collecting more informative data is more critical than the constraint violations. We control the length of this period in which the constraint-violating robots are not prioritized by the $t_W$ parameter.

## A.2  Theoretical Bounds

Here, we provide the definitions and necessary theorems to show the optimality bound of our proposed ASA and n-ASA policies. We first define submodular and monotone functions and then provide the necessary theorems to show the optimality bounds of our proposed policies.

**Definition 1** (Submodular Function). *A set function $f : 2^V \to \mathbb{R}$ is submodular for all $A \subseteq B \subseteq V$ and $e \in V \setminus B$, we have:*

$$f(A \cup \{e\}) - f(A) \geq f(B \cup \{e\}) - f(B).$$

**Definition 2** (Monotone Function). *A set function $f : 2^V \to \mathbb{R}$ is monotone if for all $A \subseteq B \subseteq V$, we have:*

$$f(A) \leq f(B).$$

**Definition 3** (Facility Location Problem). *The facility location problem is a combinatorial optimization problem where the goal is to select a subset of facilities from a set of candidate locations to minimize the cost of serving the demand points. The objective function of the facility location problem is defined as follows:*

$$f(A) = \sum_{i=1}^{n} \min_{j \in A} \hat{M}_{ij},$$

*where $A$ is the set of selected facilities, $\hat{M}_{ij}$ is the cost of serving demand point $i$ from facility $j$, and $n$ is the number of demand points.*

**Corollary 1** (Facility Location Problem is submodular for non-negative $\hat{M}_{ij}$.). *The facility location problem is submodular and monotone if the cost matrix $\hat{M}_{ij}$ is non-negative. Please refer to [50] for the detailed proof.*

Exactly following that corollary, we can show that the facility location problem presented in Equation 4 is submodular and monotone, as the cost matrix $\hat{M}_{ij}$ is non-negative and $C$ is a binary random variable.

**Definition 4** (Greedy Policy). *The greedy policy is a sequential optimization method that selects the element with the maximum marginal gain at each step. The greedy policy that selects $k$ elements is defined in Algorithm 2.*

---

**Algorithm 2** Greedy Policy

---

1: $A \leftarrow \emptyset$
2: **for** $i = 1$ to $k$ **do**
3:     $x_i \leftarrow \text{argmax}_{x \in V \setminus A} f(A \cup \{x\}) - f(A)$
4:     $A \leftarrow A \cup \{x_i\}$
5: **end for**
6: **return** $A$

---

**Theorem 1** (Adaptive Greedy Policy Optimality Bound). *Consider the adaptive greedy policy selecting elements with the maximum marginal value, conditioned on the realized value of the previously chosen elements. The approximation ratio of the adaptive greedy policy with respect to the optimal adaptive policy is $1 - \frac{1}{e}$. For the detailed proof of this theorem, please refer to [45].*

In our work, once we are able to observe whether the connectivity to robots is established or not (Alg. 1, line 8), our policy becomes adaptive. Additionally, when the $\hat{F}$ and $F$ functions match, and the marginal threshold parameter is set to 0, Theorem 1 can be applied to show the optimality bound of our proposed policies.

**Theorem 2** (Non-Adaptive Greedy Policy Optimality Bound ). *Consider the non-adaptive greedy policy that, at each step, chooses the element with the maximum marginal increase in value. The approximation ratio of this policy with respect to the optimal adaptive policy is at least $(1 - \frac{1}{e})^2$. For the detailed proof of this theorem, please refer to [45].*

Unlike ASA policy, our n-ASA policy is not able to observe whether a connection to robot $i$ has been established or not, so it is non-adaptive. Under the same conditions as the ASA policy, when the $\hat{F}$ and $F$ functions match, and the marginal threshold parameter is set to 0, Theorem 2 can be applied to show the optimality bound of our proposed n-ASA policy.

## A.3 Experimental Setups and Parameters

As stated previously, we run simulations using four different environments: ANYmal, Allegro Hand, Humanoid, and Ball Balance. Each environment has its own defined tasks, success criteria, and constraint violations. For the ANYmal robot, a constraint violation occurs when there is excessive force on the robot's knees, indicating that the robot has fallen on its knees, or when no force is exerted on the bottom of its toes, indicating that the robot has fallen on its torso. For the Ball Balance environment, a constraint violation occurs when the ball is no longer on the plate. In the Allegro Hand environment, a constraint violation happens when the cube is no longer in the robot's hand. For the Humanoid environment, a constraint violation occurs when the robot's position is below the termination height, indicating that the Humanoid has fallen down.

The definition of success is specific to each task. For instance, in locomotion tasks, success is achieved if the robot does not violate constraints and reaches a reward amount that exceeds a predefined reward threshold. For goal-specific tasks such as Ball Balance and Allegro Hand, success corresponds to reaching the goal state without violating constraints. For Ball Balance, a goal state may be one where the ball on the plate is moving within a radius smaller than the plate's radius, indicating that the robot successfully managed to control and balance the ball. For Allegro Hand, the goal state may be defined as holding the cube stable after rotating it so that the red surface faces up. That is how a single success corresponds to different achievements depending on the specific tasks assigned to each robot.

For all experiments, the key parameters are fixed and do not depend on the allocation policies: $N_{\text{human}} = 5$, $N_{\text{robot}} = 100$, $T = 10,000$ time steps, $t_R = 5$ time steps and $t_T = 5$ time steps. The hyperparameters that vary depending on the task, along with the values that yielded the best performances, are provided below in Table 2. $|S|$ and $|A|$ are the dimensionalities of the state and action spaces, respectively, $\mathcal{U}_{\text{unc}}^{\text{thres}}$ and $\mathcal{U}_{\text{risk}}^{\text{thres}}$ are the uncertainty and risk threshold values below which the uncertainty and risk are treated as zero, $t_W$ is the period during which constraints are not prioritized, allowing the robot policies to be improved by selecting informative robots rather than resetting failing robots in the first $t_W$ time steps, $threshold$ is the marginal increase threshold below which the robots are not prioritized, $\alpha_S$ is the parameter which controls the weight of the state similarity in the overall similarity measure, and $\alpha_U$ is the parameter that controls the weight of the uncertainty in the overall informativeness measure.

| **Task** | $|S|$ | $|A|$ | $\mathcal{U}_{\text{unc}}^{\text{thres}}$ | $\mathcal{U}_{\text{risk}}^{\text{thres}}$ | $t_W$ | $threshold$ | $\alpha_S$ | $\alpha_U$ |
|---|---|---|---|---|---|---|---|---|
| AllegroHand | 88 | 21 | 0.53 | 0.12 | 1250 | 0.04 | 0.37 | 0.53 |
| AnyMAL | 48 | 12 | 0.19 | 0.49 | 1000 | 0.69 | 0.72 | 0.05 |
| BallBalance | 24 | 3 | 0.47 | 0.21 | 1750 | 0.51 | 0.98 | 0.46 |
| Humanoid | 108 | 21 | 0.18 | 0.20 | 2500 | 0.23 | 0.50 | 0.10 |

Table 2: **Simulation environment hyperparameters for each task.**

## A.4 Network Configurations

Here, we explain the details of the network configurations used in our experiments. We have used four different network configurations to evaluate the adaptability of the allocation algorithms in different network conditions. Additionally, we show the connection probabilities in each network configuration in Figure 4. The network configurations are as follows:

**Always:** Always is a simple network configuration where the probability of connection to all the robots is set to 1. In this network configuration, our supervisor allocation problem is equivalent to the Interactive Fleet Learning (IFL) problem presented in [15], where the supervisor can connect to all the robots at all times.

**Mixed-Scarce:** Mixed-Scarce is a network configuration where the probability of connection to robots can be set to two different values. In this network configuration, we first divided the robots into two groups with ratios of 0.7 and 0.3. We then set the probability of connection to the robots in the first group to 0.9 and the probability of connection to the robots in the second group to 0.1. This network configuration is used to evaluate the adaptability of allocation algorithms when the connectivity to the robots is heterogeneous. Ideally, the supervisor should allocate more resources to the robots with higher connectivity to maximize the performance of the fleet.

**Ookla:** Ookla is a network configuration where the probability of connection to the robots is set based on the Ookla cellular network performance data [49]. This dataset includes the download speed, upload speed, and latency of the cellular network in different locations. We use the download speed as the metric to determine the probability of connection to the robots. We first divided the data collection points into a grid of $10 \times 10$ cells. We then calculated the average download speed of the data collection points in each cell. After that, we log-normalize the average download speed of each cell to be in the range of $[0.5, 1]$. We have set the lower bound to 0.5 to ensure that the robots in the cell with the lowest download speed have a non-zero probability of connection. We then set the probability of connection to the robots in each cell to be the normalized average download speed of the cell. This network configuration is used to evaluate the adaptability of allocation algorithms when the connectivity to the robots is based on real-world cellular network performance data, which is heterogeneous and has a more complex structure than the Mixed-Scarce network configuration.

**5G:** 5G is a network configuration where the probability of connection to the robots is set based on the real-world 5G network performance data. Please refer to Section A.8 for more details on the data collection process. The collected data was divided into 100 groups, with average latency and throughput calculated for each group and normalized to a value between 0.015 and 1. A lower bound of 0.015 ensures a non-zero connection probability for robots with the lowest throughput and highest latency. Robots in groups with throughput below 0.4 and latency above 0.6 were assigned a normalized value of 0.015. The connection probability for each group corresponds to the normalized average throughput and latency. This configuration evaluates the adaptability of allocation algorithms to realistic, heterogeneous connectivity based on real-world 5G network performance, which is more complex than other network configurations.

**Changing-Scarce:** Changing-Scarce is a network configuration where the probability of connection to robots is first set as in the Mixed-Scarce network configuration. Then, over time, the probability of connection to the robots in the first group is decreased from 0.9 to 0.1, and the probability of connection to the robots in the second group is increased from 0.1 to 0.9 linearly. This network configuration is used to evaluate the adaptability of allocation algorithms when the connectivity to the robots changes over time. Ideally, the supervisor should adapt the allocation of human supervisors to the robots based on the changes in connectivity to maximize the performance of the fleet.

## A.5   Numerical Results and Additional Metrics

In this section, we present the numerical values for all allocation policies and for all tasks under each network configuration to demonstrate that our method outperforms the baseline algorithms in all simulated scenarios, providing a novel approach to the supervisor allocation problem. We also present the percentage performance differences between our methods (ASA and n-ASA) and other methods. We present all numerical results recorded in the final timestep ($t = 10,000$) in Table 3 and the percentage differences in Figure 5.

In addition to the presented metrics in the main paper, we also provide the following metrics to evaluate the performance of our method against the baselines: (1) cumulative human actions, which measures the total number of time-steps humans supervised the robots, (2) cumulative idle time, which is the total duration that robots remain in the constraint violating states while awaiting hard resets, (3) cumulative hard resets, which records the total number of hard resets performed by the human supervisors, (4) cumulative reward, which is the total reward accumulated by the all robots in the fleet, and (5) cumulative success, which is the total number of tasks successfully completed by the entire fleet. We present the these metrics in Figures 6, 7, 8, 9, and 10 respectively.

In Figures 6, 7 and 8, we can observe that the ASA and n-ASA methods use fewer human actions and are able to reduce idle time and hard resets compared to the other benchmarks. This is due to the fact that our method prioritizes the robots with higher connectivity to reset and supervise, wasting less human resources on robots with low connectivity, unlike the other method possibly trying to reset the robots with low connectivity. Additionally, in Figures 9 and 10, we can see that our ASA and n-ASA methods are among the top-performing methods in terms of cumulative reward and cumulative success in all network configurations and tasks, proving that our method is much more efficient in terms of human resource utilization.

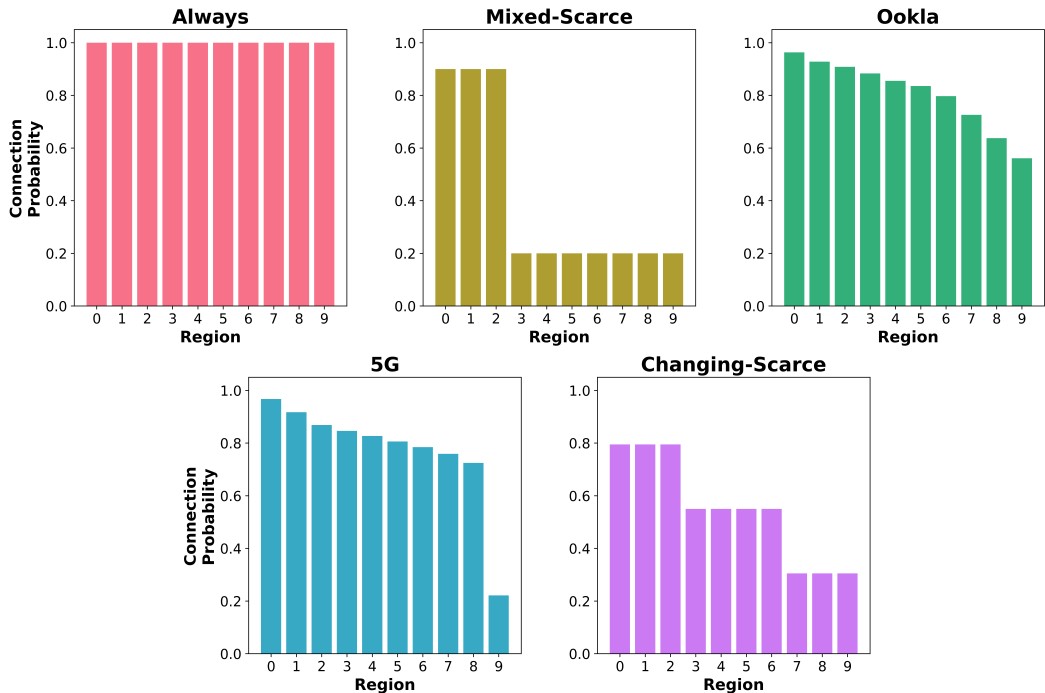

Figure 4: **Connection probabilities for each network configuration.** This figure shows the connection probabilities of the robots in the fleet for each network configuration. For easier visualization, we have grouped 100 robots into 10 groups of 10 robots each and presented the average connection probability for each group. As we go from the Always network to the Changing-Scarce network, the connection probabilities of the robots get more heterogeneous. This heterogeneity in the connection probabilities is crucial for evaluating the adaptability of the allocation algorithms in different network configurations.

## A.6  Ablation Studies

In this section, we conduct further experiments using our adaptive submodular allocation, ASA, policy to explore the following: (1) the sensitivity of the system to the ratio of the number of robots $N_{\text{robot}}$ to the number of humans $N_{\text{human}}$ (Figure 11), (2) the impact of varying the minimum intervention time $t_T$ (Figure 12), and (3) the impact of changing the hard reset time $t_R$ (Figure 13). Each experiment is averaged over three different random seeds, and the shaded regions correspond to one standard deviation. We plotted four different metrics: (1) cumulative success, (2) RoHE, (3) cumulative hard resets, and (4) cumulative idle time. Cumulative hard resets represent the total number of hard resets performed by human supervisors when the robots violate constraints. Cumulative idle time is the total time, in time steps, that robots remain idle while waiting for a hard reset.

**Number of Human Supervisors:** We tested the ASA policy to evaluate its sensitivity to different numbers of human supervisors (Fig. 11). Keeping the number of robots constant, we simulated scenarios with 1, 5, 10, 25, and 50 human supervisors. In all simulated tasks, as the number of human supervisors increases, idle time decreases because more human resources are available, resulting in shorter idle periods before robots are teleoperated and reset. However, despite the cumulative success values rising with more supervisors, the RoHE values tend to decrease. This happens because allocating more humans doesn't always lead to a higher return on human effort. The most informative and important robots are already being selected, so adding more supervisors doesn't necessarily result in a significant marginal gain. Therefore, a low number of human supervisors is insufficient as robots remain idle for long periods and violate constraints more frequently, while a large number of supervisors creates a surplus and decreases efficiency.

**Minimum Intervention Time:** While keeping other hyperparameters fixed, we varied the minimum intervention time and ran our policy. We observed that when the minimum intervention time is long, such as 100 or 500 time steps, the robot fleet performance significantly decreases. This is because human supervisors spend a lot of time teleoperating a single robot, which results in lower RoHE and cumulative success values, and a substantial increase in idle time. Conversely, when the minimum intervention time is very short, such as 1 time step, performance improves in terms of both RoHE and cumulative success for most tasks. This is because each human supervisor spends less time on a single robot and can attend to more robots within 10,000 time steps, thus enhancing overall fleet performance as the minimum intervention time decreases.

| NETWORK | ALLOCATION POLICY | ALLEGROHAND | | ANYMAL | | BALLBALANCE | | HUMANOID | |
|---|---|---|---|---|---|---|---|---|---|
| | | RoHE | CUMULATIVE SUCCESS | RoHE | CUMULATIVE SUCCESS | RoHE | CUMULATIVE SUCCESS | RoHE | CUMULATIVE SUCCESS |
| ALWAYS | RANDOM | 10.48 | 649.7 | 2.18 | 160.0 | 3.61 | 1796.0 | 0.01 | 1.67 |
| | FT | 11.10 | 2751.0 | - | - | 2.87 | 1437.0 | - | - |
| | FE | 10.03 | 2021.67 | 2.81 | 127.33 | 6.82 | 1185.0 | 1.04 | 296.67 |
| | FD | 13.48 | 3352.33 | 3.12 | 189 | 7.92 | 1499.67 | 1.93 | 424.67 |
| | N-FD | 13.48 | 3352.33 | 3.12 | 189 | 7.92 | 1499.67 | 1.93 | 424.67 |
| | N-ASA (OURS) | **16.30** | **5094.67** | **3.76** | **246.33** | 10.67 | **1802.67** | 2.22 | 503.33 |
| | ASA (OURS) | 14.87 | 5064.33 | 3.41 | 241.0 | **10.87** | 1785.67 | **2.28** | **530.0** |
| MIXED-SCARCE | RANDOM | 3.39 | 302.0 | 0.65 | 79.33 | 2.64 | 1319.0 | 0.00 | 0.67 |
| | FT | 4.05 | 1713.0 | - | - | 1.98 | 988.0 | - | - |
| | FE | 3.69 | 1616.33 | 0.89 | 110.33 | 3.54 | 1327.67 | 0.09 | 45.0 |
| | FD | 4.94 | 2031.0 | 1.29 | 180 | 3.32 | 1253.33 | 0.33 | 159.33 |
| | N-FD | 5.86 | 2370.66 | 0.27 | 137.66 | 1.07 | 534.33 | 0.37 | 184.67 |
| | N-ASA (OURS) | 8.15 | **3850.67** | **2.46** | **244.33** | 5.85 | 1580.67 | **1.95** | **499.67** |
| | ASA (OURS) | **8.16** | 3817.33 | 2.29 | 235.33 | **7.27** | **1621.0** | 1.85 | 491.0 |
| OOKLA | RANDOM | 8.45 | 576.33 | 1.97 | 164.67 | 3.57 | 1782.33 | 0.01 | 1.67 |
| | FT | 9.37 | 2583.33 | - | - | 2.77 | 1383.33 | - | - |
| | FE | 8.34 | 2072.33 | 2.22 | 126.0 | 7.46 | 1301.33 | 0.70 | 235.33 |
| | FD | 10.65 | 2986.33 | 2.76 | 186.67 | 6.45 | 1443.33 | 1.47 | 392.33 |
| | N-FD | 12.38 | **6191.33** | 0.55 | 275.33 | 2.98 | 1491.67 | 1.03 | **517.33** |
| | N-ASA(OURS) | **13.99** | 4694.33 | **3.20** | 240.0 | 10.96 | **1791.67** | **2.25** | 506.67 |
| | ASA (OURS) | 13.75 | 4936.0 | 3.11 | 225.67 | **11.41** | 1741.33 | 2.15 | 510.33 |
| 5G | RANDOM | 2.47 | 228.67 | 0.36 | 47.33 | 2.81 | 1401.33 | 0.00 | 0.00 |
| | FT | 4.75 | 2131.67 | - | - | 0.69 | 344.67 | - | - |
| | FE | 3.98 | 1789.67 | 1.07 | 137.0 | 3.01 | 1315.0 | 0.37 | 169.33 |
| | FD | 5.40 | 2213.0 | 1.40 | 200 | 2.75 | 1209.0 | 0.79 | 319.33 |
| | N-FD | **12.01** | **6009** | 0.52 | **259.33** | 2.86 | 1433 | 0.96 | **479.67** |
| | N-ASA (OURS) | 7.66 | 3705.33 | 1.61 | 246.33 | **7.45** | **1758.33** | 1.44 | 444.33 |
| | ASA (OURS) | 7.43 | 3658.00 | **1.63** | 239.33 | 6.39 | 1703.33 | **1.51** | 470.0 |
| CHANGING-SCARCE | RANDOM | 1.9 | 183 | 0.08 | 12 | 1.89 | 945 | 0 | 0 |
| | FT | 3.17 | 1418.66 | - | - | 0.54 | 270.66 | - | - |
| | FE | 2.70 | 1222.33 | 0.21 | 76.67 | 1.48 | 637 | 0.01 | 5 |
| | FD | 1.66 | 672.66 | 0.27 | 102.33 | 0.98 | 470.66 | 0.07 | 35.33 |
| | N-FD | 1.39 | 695.66 | 0.23 | 115.66 | 0.43 | 219 | 0.07 | 39 |
| | N-ASA(OURS) | 2.65 | 1302.64 | 0.52 | 185.34 | 1.51 | 661.33 | 0.19 | 86.33 |
| | ASA (OURS) | **12.23** | **3612.33** | **1.04** | **230** | **4.81** | **1328.33** | **1.01** | **391.66** |

Table 3: **Numerical results for all network and task simulations.** We present RoHE and cumulative success values in the final timestep ($t = 10,000$) for 4 different environments under 4 different network configurations.

**Hard Reset Time:** Finally, we ran the ASA policy with different hard reset times. We observed that as the hard reset time increases, the fleet performance decreases. This is because it takes longer for human supervisors to reset the robots, resulting in fewer hard resets within 10,000 time steps. Consequently, the idle time increases, reducing the overall performance of the robot fleet.

## A.7 Complexity Analysis and Optimality Bounds for Allocation Algorithms

Here, we present the complexity analysis and optimality bounds for our allocation algorithms.

### A.7.1 Complexity Analysis

We now explain the complexity of our allocation algorithms in terms of the number of robots $N_{robot}$ and the number of humans $N_{human}$ in the system and function evaluations. As we have discussed in Section 4, our allocation algorithm is based on a greedy algorithm that selects the robots based on the stochastic submodular maximization objective. It is a well-known result that the number of function evaluations for the greedy algorithm is $O(N_{robot}N_{human})$. As both of our algorithms are based on the greedy algorithm, the computational and time complexities of our algorithms, ASA and n-ASA, are both $O(N_{robot}N_{human})$.

## A.8 Real World 5G Network Data

In addition to the simulated network connectivity data, we also evaluate our allocation policies using real-world 5G network connectivity data collected via setup detailed in [51]. This setup includes two hardware components: a mobile edge device and a local server. The edge device, which can be a robot, a mobile phone, or a computer, acts as the connection client. The local server functions as the cloud. In our scenario, we consider the edge device to be the robot and the local server to be the cloud or the server from which human supervisors connect to the robots. The local server sends packets to the edge device, and the edge device responds with packets to confirm receipt. During this process, the local server calculates latency and throughput, saving this data to a local file. This continues for a predetermined data collection period of 24 hours. Two key aspects of this setup are: (1) the edge device is connected to a 5G cellular network, specifically 5G cellular provided by AT&T, and (2) it is mobile. This allows us to collect data anywhere, whether moving or stationary, for any desired period. To obtain data that realistically simulates human teleoperation connectivity, we collected data in a building where actual teleoperation and robotic tasks are conducted. After collecting the data, we divided and

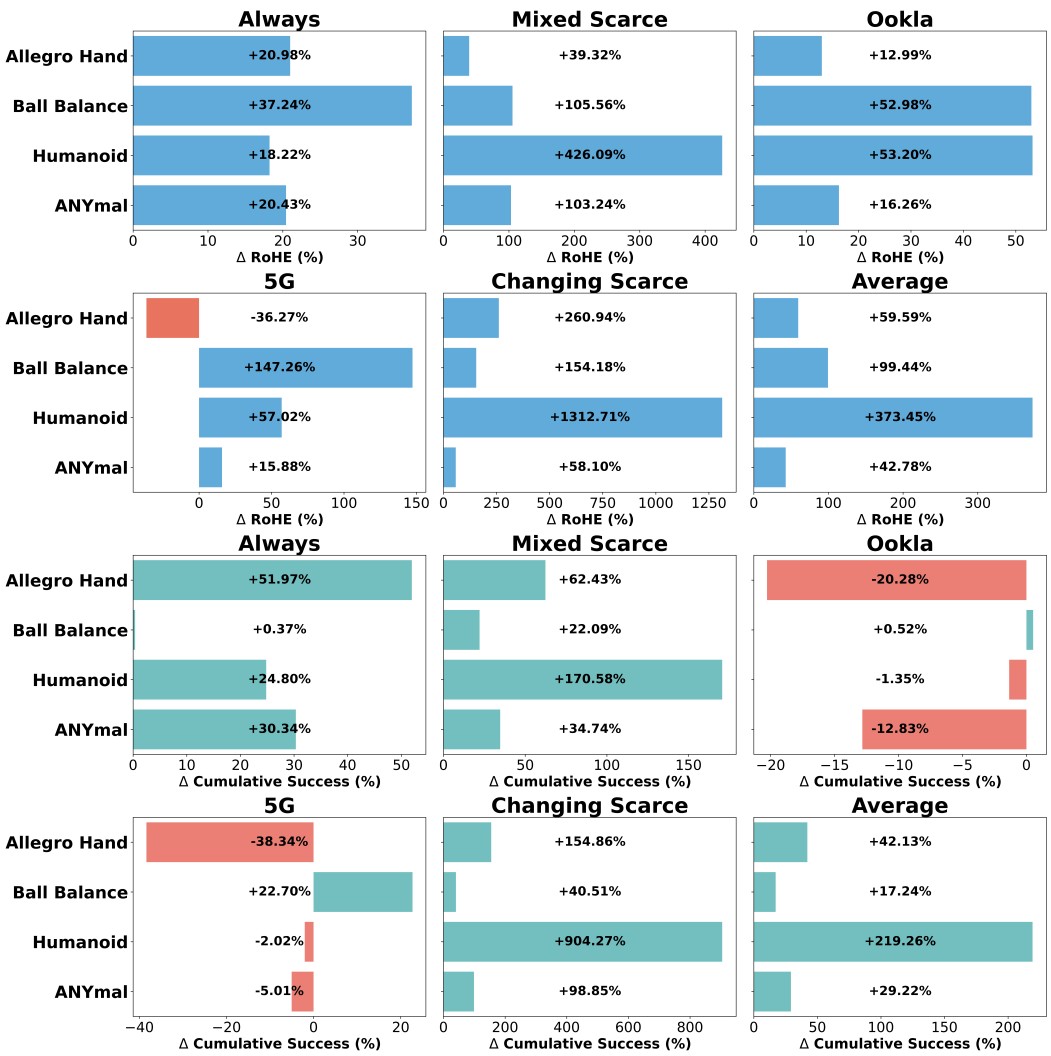

Figure 5: **Performance gains of ASA and n-ASA policies over baseline methods across different environments and network combinations.** The top two rows show the percentage difference in RoHE, while the bottom two rows depict the percentage difference in cumulative success compared to the best baseline methods. Notably, as highlighted in the bottom right figure, our policies outperform the baseline methods, particularly when averaged across different network combinations, demonstrating the robustness and effectiveness of our approach.

clustered it into 100 different groups. This division helps correlate the data with our fleet learning simulation environment, which has 100 robots in different locations. For each group, we calculated the average latency and throughput. We normalized the average values between 0 and 1, such that groups with high latency and low throughput values have a connection probability closer to 0, and groups with low latency and high throughput values have a connection probability closer to 1. Now that we have 100 different connection probability values, we randomly assigned them to 100 simulated robots. We illustrate the data collection setup in Figure 14. Additionally, we present the average throughput and latency for each group in Figure 15.

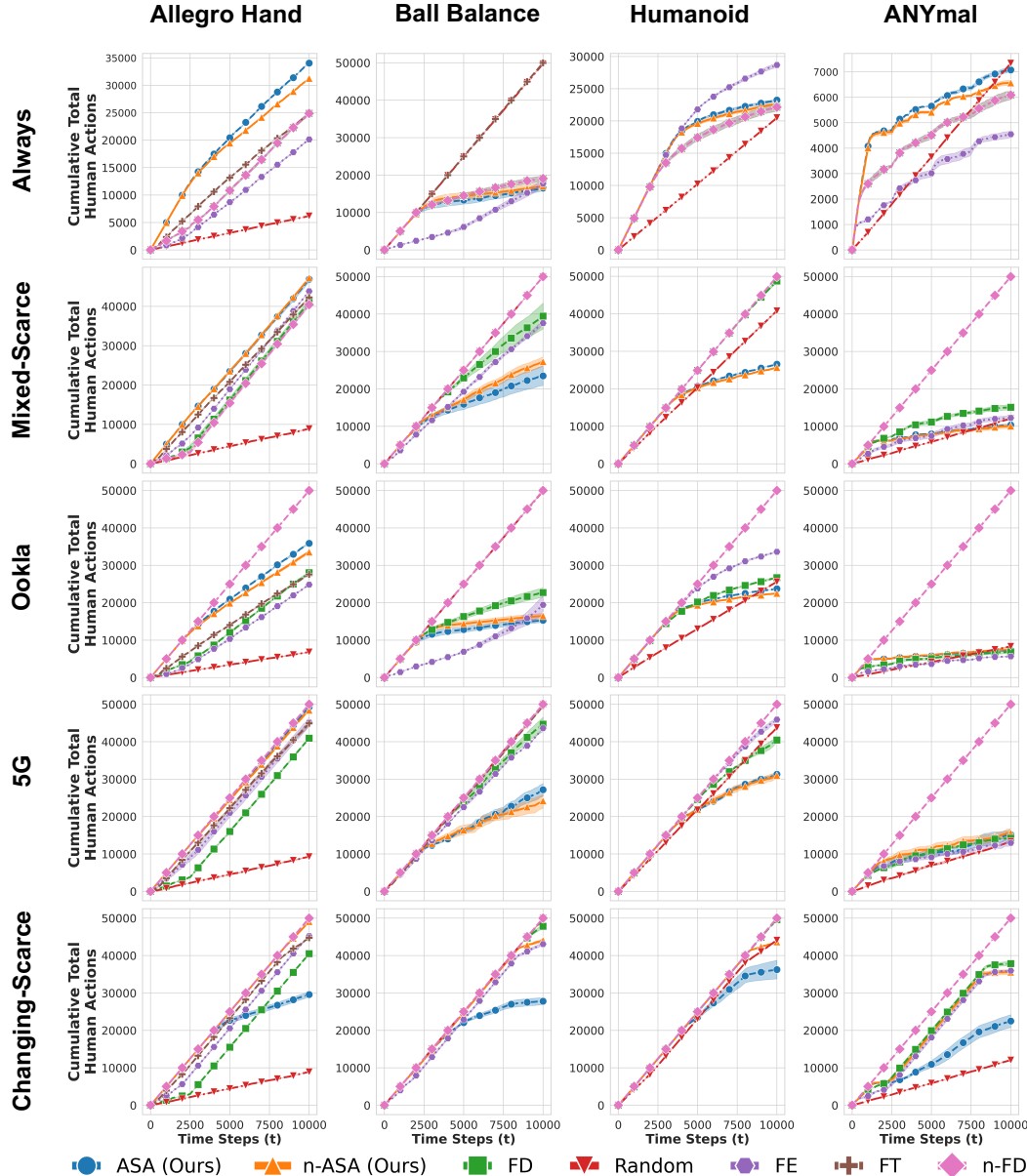

Figure 6: **Cumulative human actions for different network configurations in different environments.** This figure shows cumulative human actions, which is the sum of the total number of teleoperation/supervision attempts of the human supervisors, for different network configurations in different environments. The x-axis represents the time steps, and the y-axis represents the cumulative human actions.

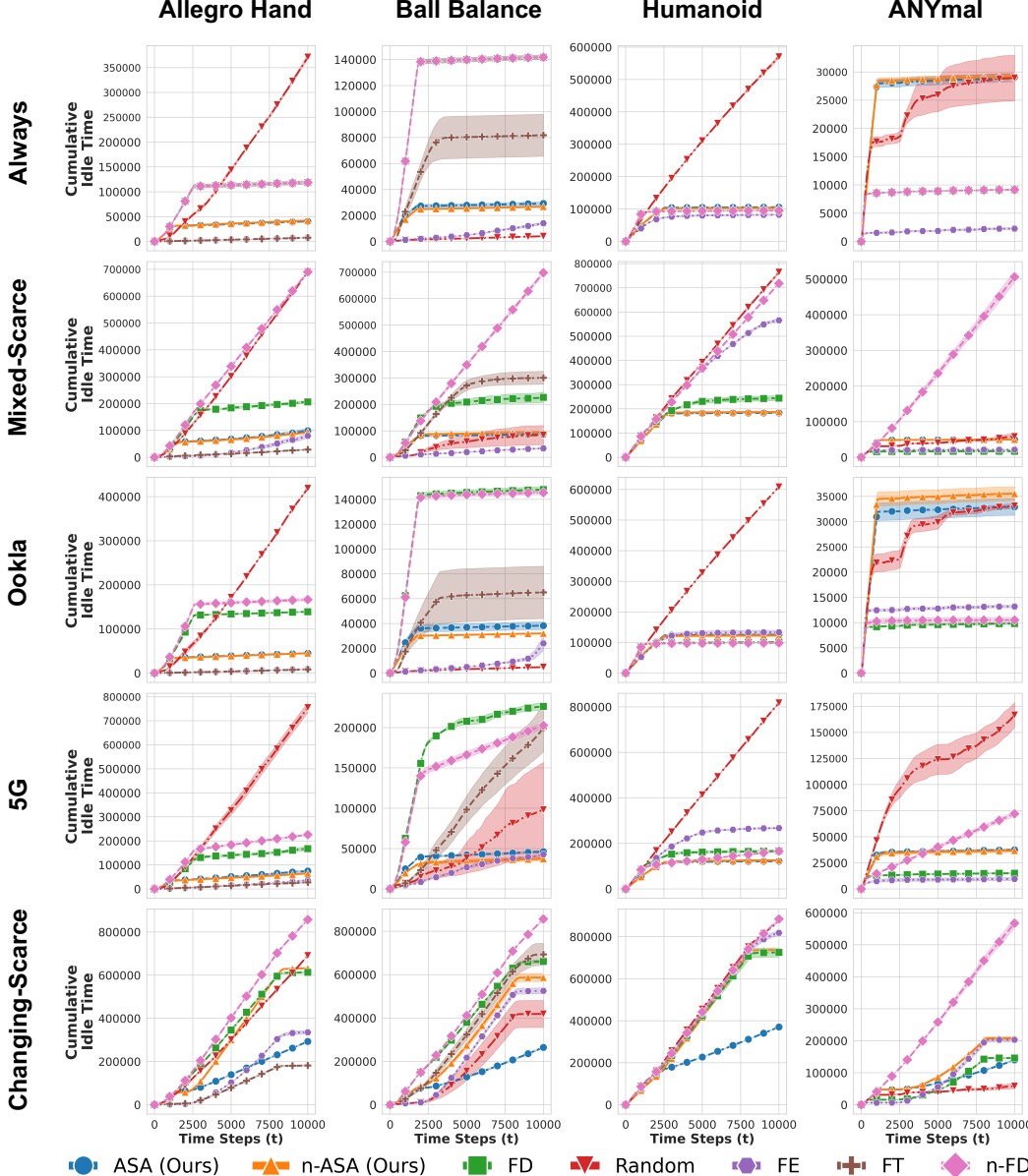

Figure 7: **Cumulative idle time for different network configurations in different environments.** The plots show the cumulative idle time, which is the sum of the total duration that robots remain in the constraint-violating states while awaiting hard resets, in different environments and network configurations. The x-axis represents the time steps, and the y-axis represents the cumulative idle time.

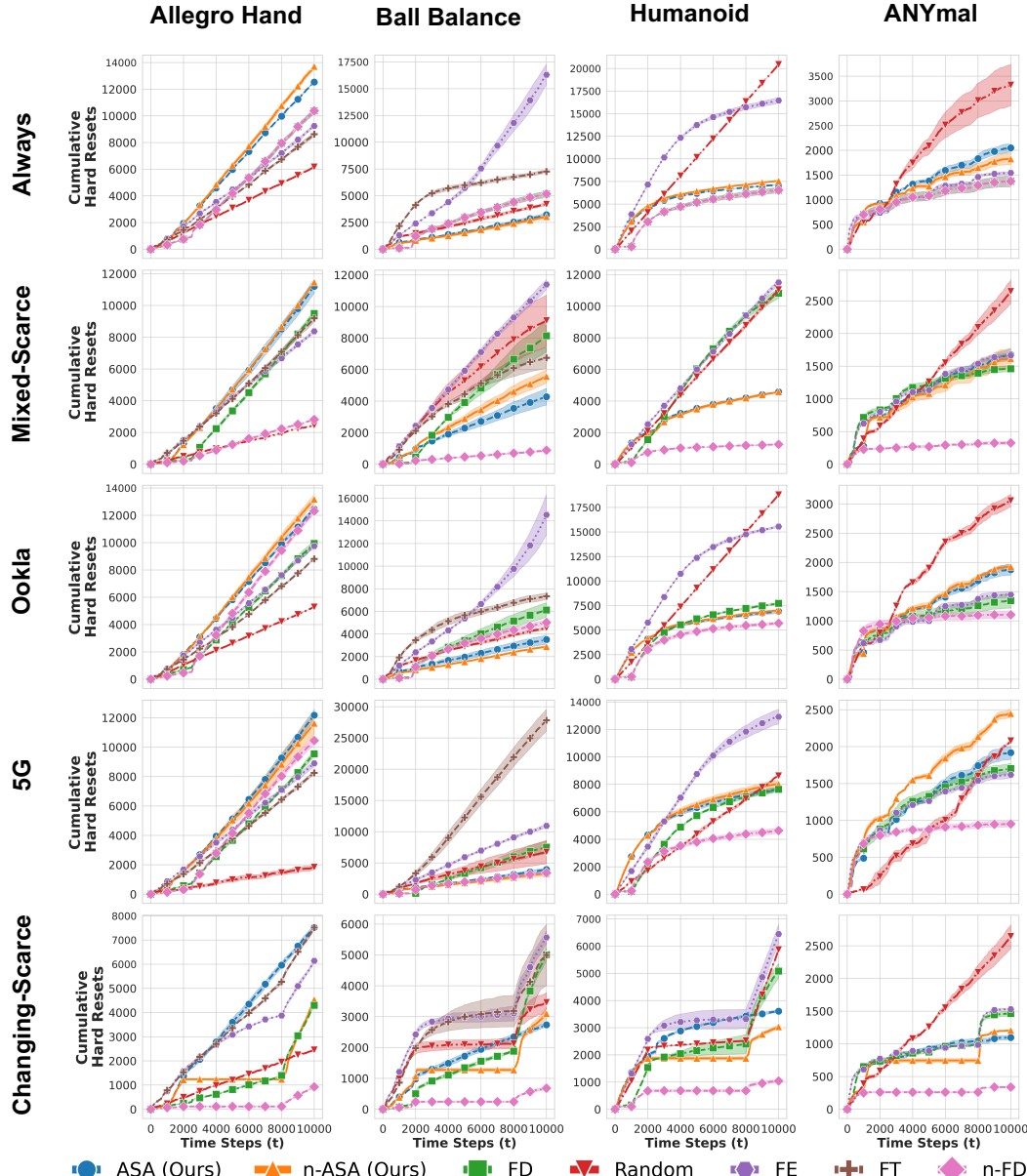

Figure 8: **Cumulative hard resets for different network configurations in different environments.** The plots show the total number of hard resets performed by the human supervisors in different environments and network configurations. The x-axis represents the time steps, and the y-axis represents the cumulative hard resets.

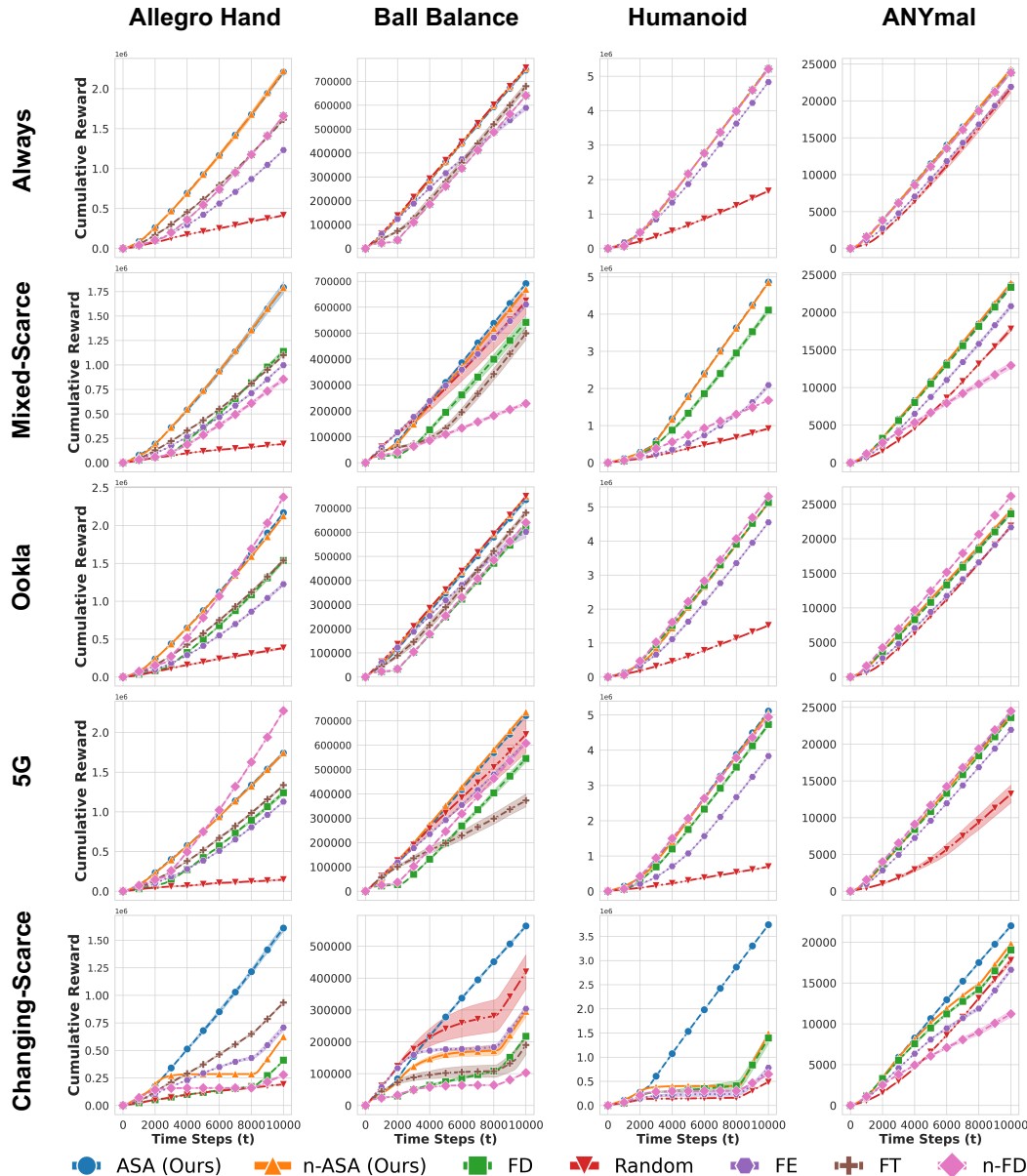

Figure 9: **Cumulative reward for different network configurations in different environments.** The plots show the cumulative reward, which is the sum of the total reward accumulated by all robots in the fleet, for different network configurations in different environments. The x-axis represents the time steps, and the y-axis represents the cumulative reward.

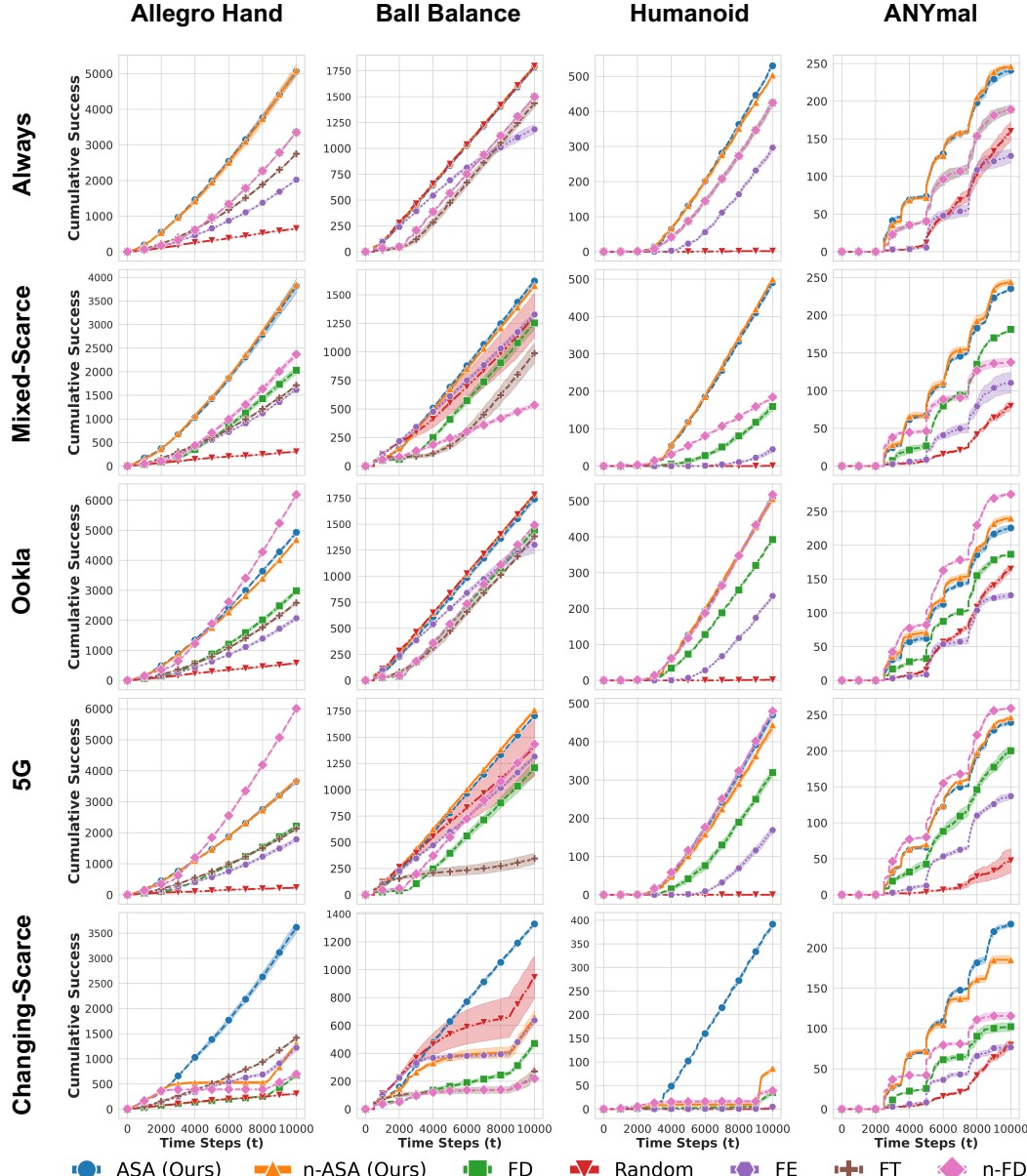

Figure 10: **Cumulative success for different network configurations in different environments.** The plots show the cumulative success, which is the sum of the total number of successful tasks completed by all robots in the fleet, for different network configurations in different environments. The x-axis represents the time steps, and the y-axis represents the cumulative success.

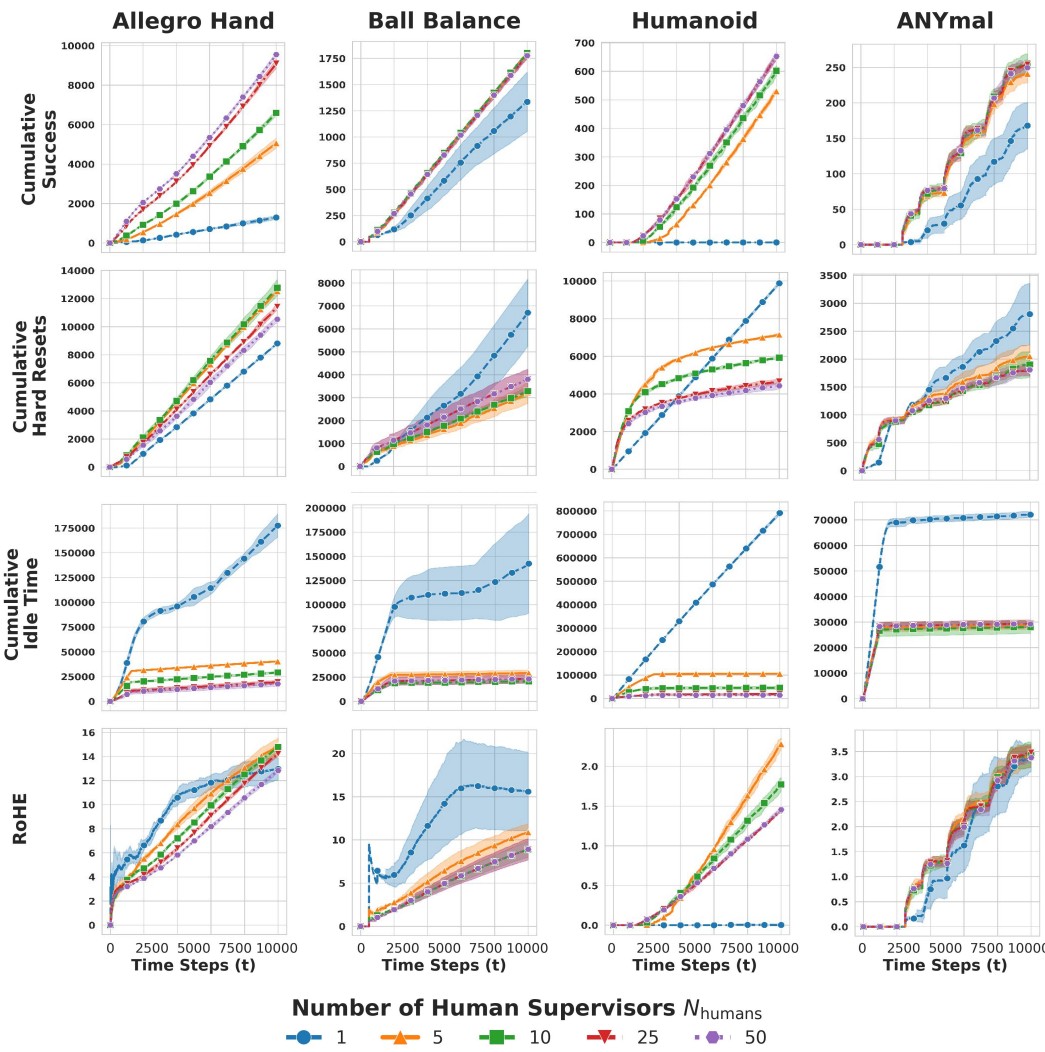

Figure 11: **Simulation results with varying numbers of human supervisors.** The figure shows the simulation results of the ASA policy for each task under the Always network configuration, with the number of robots $N_{robot}$ fixed at 100, while the number of human supervisors $N_{human}$ varies.

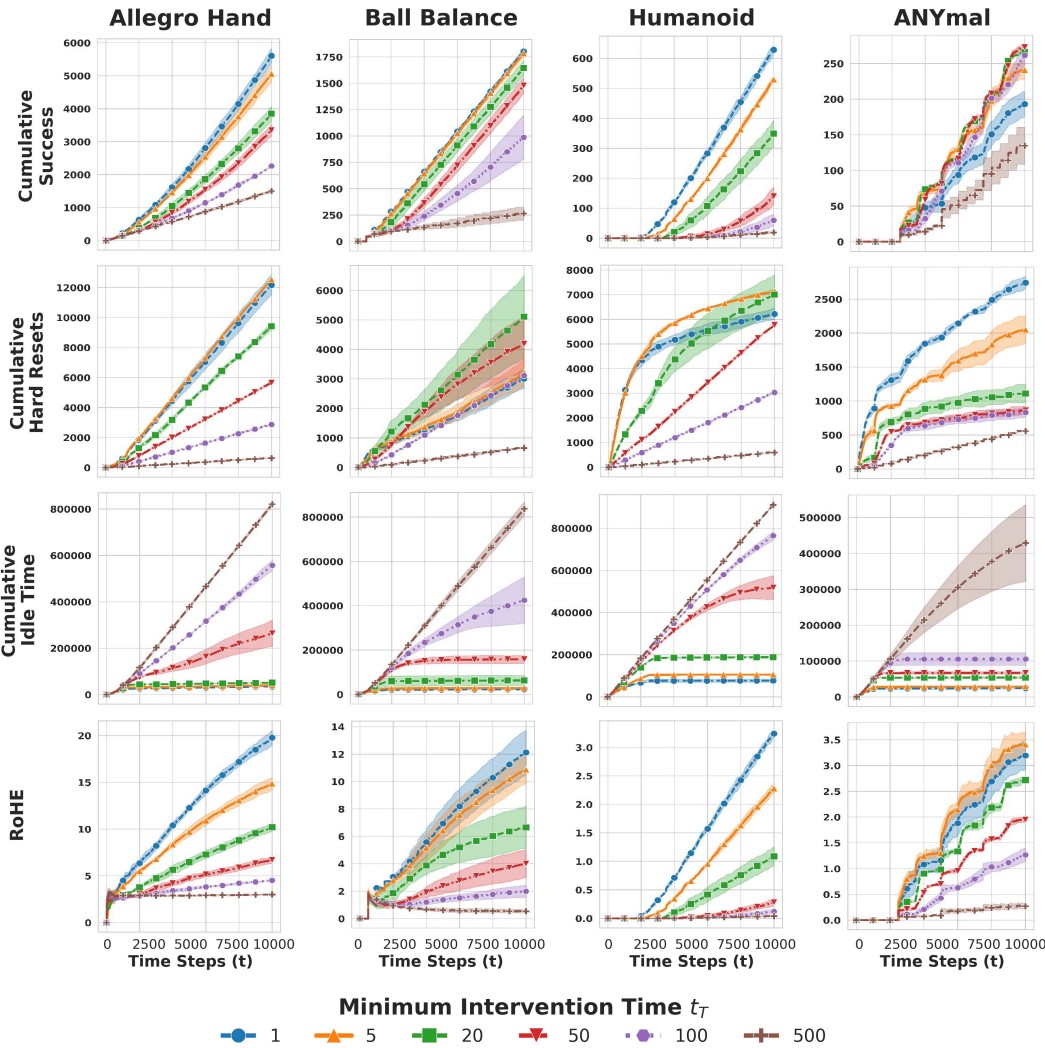

Figure 12: **Simulation results with varying minimum intervention times.** The figure shows the simulation results of the ASA policy for each task under the Always network configuration over $T = 10,000$ time steps, with the number of robots $N_{\text{robot}}$ fixed at 100, the number of human supervisors $N_{\text{human}}$ fixed at 5, and the minimum intervention time $t_T$ varying.

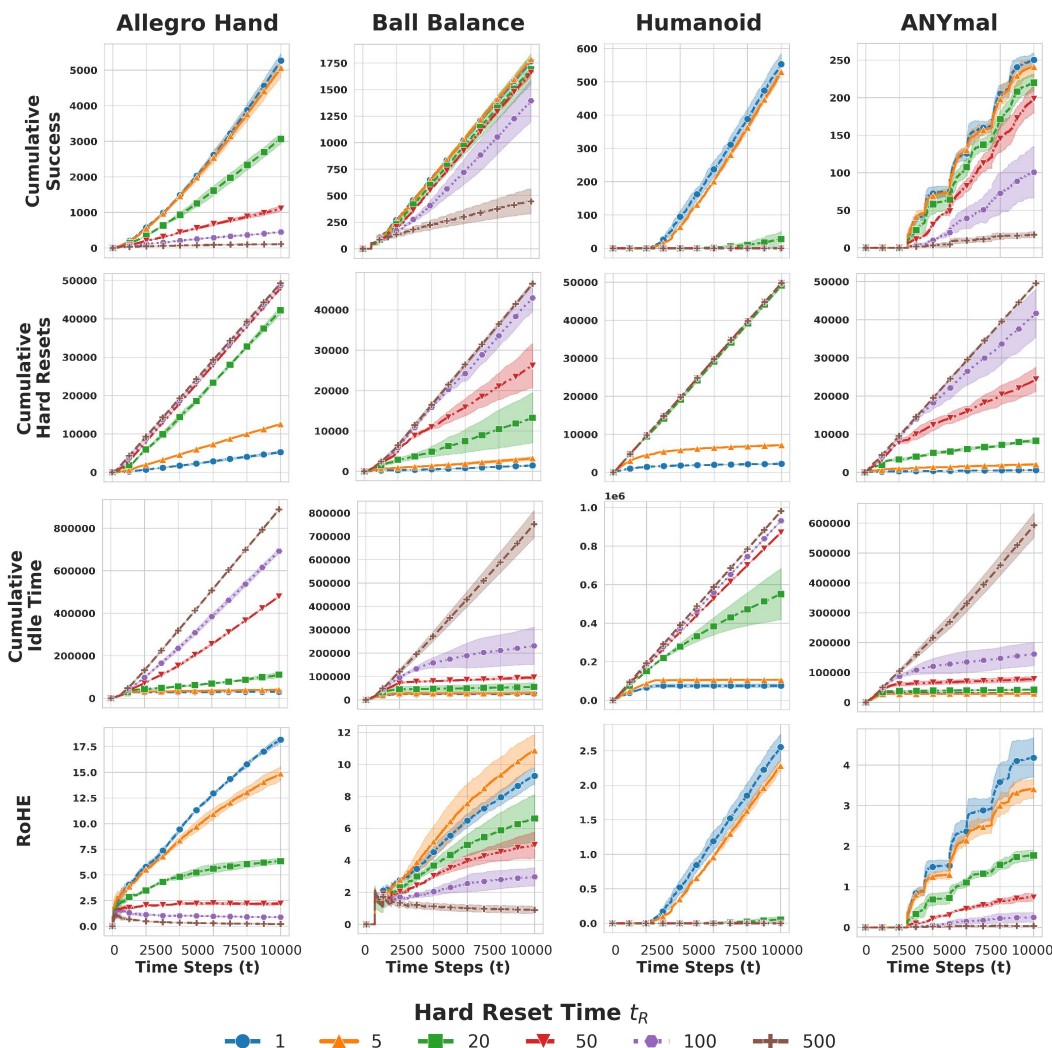

Figure 13: **Simulation results with varying hard reset times.** The figure shows the simulation results of the ASA policy for each task under the Always network configuration, with the number of robots $N_{\text{robot}}$ fixed at 100, the number of human supervisors $N_{\text{human}}$ fixed at 5, and the hard reset time $t_R$ varying.

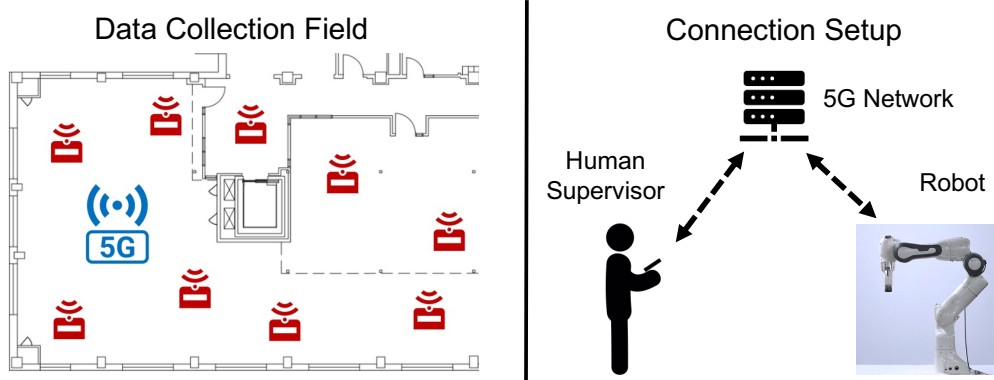

Figure 14: **Data collection setup for 5G network.** We collected 5G network connectivity data from a real-world robotics laboratory floor. An example of the floor plan is shown in the left figure. In this floor plan, the red devices represent the locations of the robots on the laboratory floor. Our data collection setup is shown in the right figure. For each location, we established a connection between a human supervisor using a 5G-enabled smartphone and a robot server through a 5G base station and a 5G modem. We have collected various network parameters including throughput, latency, and signal strength for each location. We then processed this data to obtain the network connectivity information for the robots in our experiments.

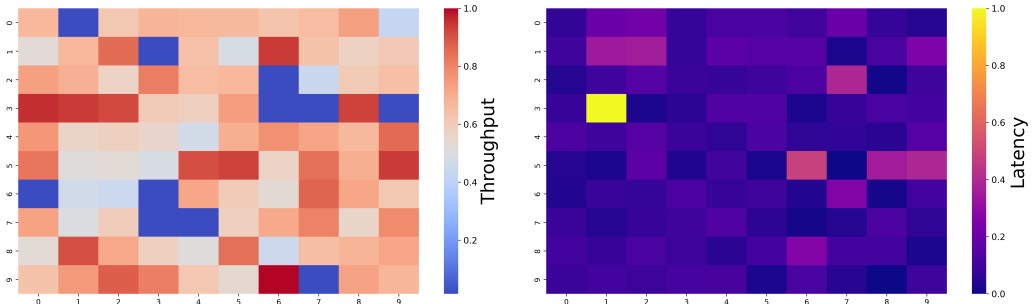

Figure 15: **5G network performance metrics.** This figure shows the key performance metrics of the 5G network data collected from the real world. Here, on the left, we show the average throughput for each group of robots. The throughput is normalized between 0 and 1, where 0 represents low throughput and 1 represents high throughput. On the right, we show the average latency for each group of robots, where the latency is normalized between 0 and 1, where 0 represents low latency, and 1 represents high latency. We then use these metrics to determine the probability of connection to the robots in our experiments.

