# OpenReview forum: "Fleet Supervisor Allocation: A Submodular Maximization Approach"
_robot-learning.org/CoRL/2024/Conference — CoRL 2024_

### Official Review · Reviewer_ZFVB · 2024-07-16
**Interesting research problem, preliminary results**

**Originality:** 3
**Technical Quality:** 3
**Clarity Of Presentation:** 3
**Potential Impact:** 2
**Recommendation:** 2
**Confidence:** 4

**Review:**

This paper considers an interesting and important problem. The current definition, however, is missing the temporal aspect. The problem is reduced to taking a snapshot of the current state, and running a static allocation algorithm. While it is true that greedily optimizing a submodular function offers theoretical guarantees, these only apply to the static version of the problem only. No reasoning is performed over what might happen during subsequent steps, which limits the novelty of the work. However, please note that I am not suggesting that this is not a valid method (the experimental results suggest that it is, and as a practitioner this is the first thing I would have tried), but as a reader of a conference paper I am left wondering if something better could be done be planning over multiple allocation steps.

Regarding the temporal aspects, I believe the binary connectivity model could be further generalized to account, for example, the need for humans to spend non-negligible time for context switching, or unexpected loss of connectivity while in the middle of a task (it is not clear if the current setting connectivity might drop during an intervention). In summary, I would classify this as an experimental paper with some interesting preliminary results.

I believe there are a few points requiring further clarifications (see my questions for rebuttal). A very interesting (in my opinion) scenario - self-driving cars - is missing from the experimental evaluation. It would have been nice to have it.

Minor comments:

1. Consider using R to denote the set of robots and 2^R to denote all the possible subsets (instead of the current X, which does not seem to be used consistently (see the paragraph before Eq. (4)).

**Quality Of The Limitations Section:**

2

**Questions For Rebuttal:**

1. Could you please elaborate on the notion of "hard reset time" which was set to 5 time steps? How could a smaller / larger value impact an experiment?
2. For each intervention, how is the duration determined during an experiment (in addition to having a minimum time of 5 steps)? Why is the duration not a part of the problem statement / output of the allocation algorithm?
3. During an experiment, what happens when the robot is not being supervised? Does it keep following its current policy? Is that deterministic or random? How is the data collected in those steps used?
4. Did you consider integrating the temporal aspect in the allocation algorithm? If so, what difficulties did you encounter?

**Robotics Focus:**

3

**Summary Of Paper:**

This paper considers the problem of human supervision allocation in multi-robot settings with uncertain connectivity. The proposed method is based on the design of a submodular allocation function and its greedy optimization. Simulations are used to investigate the impact of connectivity quality across different domains and against different baselines.

**Summary Of Recommendation:**

This paper has potential but I believe this work does not get to the core of the problem.

---

### Official Review · Reviewer_rf9j · 2024-07-19
**Submodular Maximization for the Supervisor Allocation Problem**

**Originality:** 3
**Technical Quality:** 3
**Clarity Of Presentation:** 3
**Potential Impact:** 2
**Recommendation:** 3
**Confidence:** 3

**Review:**

This work builds on the formulation and algorithm of the FleetDAgger solution to the robot supervisor allocation problem.
The supervisor allocation problem is extended to include variable connectivity between supervisors and robots, with each robot having a fixed probability of a failed connection. The authors then introduce a modular objective for the value of supervision for each robot.
This formulation is the most interesting and original portion of the paper, and it captures the similarity across robots, and safety and informativeness of each robot. The use of the greedy solution in Algorithm 1 follows directly from the use of the modular objective in the maximization problem. The only difference from standard submodular maximizaiton is the marginal threshold parameter.

The experiments are thorough and interesting. Real 5G data was collected from robots in the lab to improve the realism of the simulated experiments. Also, the authors implement several Fleet versions of multiple state-of-the-art baselines, and show their algorithm's improvement over these baselines. The experimental results are all in simulation, but are very thorough and use multiple simulated robotic domains. There is a lack of discussion about why the adaptive ASA approach vs. n-ASA doesn't make a big difference. My opinion is that it has to do with the formulation of the connectivity problem having constant probability of dropped connections.

**Quality Of The Limitations Section:**

2

**Questions For Rebuttal:**

Would it be possible to highlight the differences between ASA and n-ASA in Algorithm 1?

Please provide additional discussion of [46]. How is Algorithm 1 derived from existing work and how are the bounds derived at the end of section 4.2?

Please discuss the marginal threshold parameter $\Delta$ and how might that be chosen in practice.

I would like to see more discussion of Eq. 5 to explain the $U(i)$ and $S(j,i)$ functions and how they were selected for the experiments.
Specifically, Figure 4 shows that ASA and n-ASA outperform all other baselines when connectivity is stable, which is an interesting result. Can you provide more numerical detail on why these methods outperform the baselines?

The assumption that robots' connectivity probabilities $C$ are known a priori and are constant limits the applicability of this approach to mobile robotics. Please explain how this affects the relative performance of n-ASA vs. ASA. You may want to add this analysis to expand the Limitations section.

A few other corrections:
- Ensure that the acronym ASA is defined prior to its first use in the abstract, and capitalized in text.
- Table 1 typo: N-ASSA and ASSA should be N-ASA and ASA

**Robotics Focus:**

2

**Summary Of Paper:**

This work uses submodular maximization to solve the supervisor allocation problem under network connectivity constraints.

**Summary Of Recommendation:**

The work is incremental and has no robot experiments. Its formulation limits its use in mobile robotics, where connectivity concerns would be most relevant.

---

### Official Review · Reviewer_hzFg · 2024-07-20
**Interesting extension of work on Robot Fleet Supervision.  Weak Accept:    Strong Accept if questions are addressed in rebuttal.**

**Originality:** 3
**Technical Quality:** 4
**Clarity Of Presentation:** 4
**Potential Impact:** 3
**Recommendation:** 4
**Confidence:** 3

**Review:**

The paper addresses an important and timely challenge for training and managing real-world robot fleets. To my knowledge, it is the first to study multi-robot interactive imitation learning with variable network connectivity. The proposed approach is intuitive and the use of submodularity is interesting.   the manuscript is written clearly. The empirical results are convincing, with evaluation over different realistic network configurations and a demonstrated advantage over existing state-of-the-art techniques.

The detailed Appendix is very helpful!

However, to be accepted it is important to address some weaknesses.

First, the evaluation is limited. All the simulation results are limited to NVIDIA Isaac Gym environments. Figure 3, Figure 4, and Table 1 all show results from the same set of experiments rather than different ones. As such, it is unclear whether the authors’ findings extend beyond Isaac Gym to other domains or physical robots.  Also, As the authors note, while there are simulation experiments with real-world datasets, there is no evaluation with physical hardware.   The paper would be greatly improved with simulation in another environment and ideally on real hardware although the latter is not required.

Second, the  problem formulation is very similar to that of the Fleet-DAgger work it extends, the only addition being connection probabilities. The contributions and new novel aspects with respect to the prior work should be explicitly mentioned in the text. Also, the connection probabilities appear to be assigned statically to each individual robot; should they not vary based on the current location of each robot or otherwise depend on the state? Some clarification is needed here. Moreover, the value of supervision provided in Equation 5 is said to allow modular definitions for each component, but the text does not specify how these are defined for the ASA policy. Without clarification, these components look to be similar to the baselines: U(i) and C(i) appear to be the uncertainty and constraint violation metrics from the Fleet-DAgger work, and it is unclear if the proposed S(i) similarity metrics such as Euclidean distance can scale to image-based policies.  The pape would be improved if these can be addressed in the rebuttal.

Third, the previous Fleet-DAgger baselines appear to be evaluated without modification in the variable network connectivity setting of this work. These baselines should not be expected to perform as well since they have no knowledge of the robot connectivity. A more appropriate choice of baseline would be a heuristic that adapts the Fleet-DAgger algorithms to this setting: for instance, one that thresholds on the connection probabilities before allocating supervisors with Fleet-DAgger. The paper could be improved with additional experiment, especially since the performance benefit of ASA over Fleet-DAgger is small for the fully connected network in Figure 3.

**Quality Of The Limitations Section:**

2

**Questions For Rebuttal:**

Please respond to the points raised in the main review.  Is it possible to perform additional experiments?

In addition to the clarifications requested in the main review, why does N-ASA often outperform ASA in Table 1? Should ASA not be strictly better than N-ASA, given its additional observability?

Also, please expand the Limitations section with more details, including those above...

**Robotics Focus:**

3

**Summary Of Paper:**

This paper extends the interactive fleet learning problem formulation proposed by Hoque et al. to model uncertainty over network connectivity between robots and their human supervisors. The paper presents a new algorithm for human-to-robot supervisor allocation called Adaptive Submodular Allocation (ASA) based on the theory of submodular maximization. Simulation results suggest ASA outperforms existing baselines in terms of cumulative successes and return on human effort in a range of network configurations.

**Summary Of Recommendation:**

Interesting extension of work on Robot Fleet Supervision.  The paper addresses an important and timely challenge for training and managing real-world robot fleets. To my knowledge, it is the first to study multi-robot interactive imitation learning with variable network connectivity. The proposed approach is intuitive and the use of submodularity is interesting.   the manuscript is written clearly. The empirical results are convincing, with evaluation over different realistic network configurations and a demonstrated advantage over existing state-of-the-art techniques.   However, to be accepted it is important to address some weaknesses.  Weak Accept:    Strong Accept if questions are addressed in rebuttal.

---

### Author Rebuttal · Authors · 2024-08-11

We thank each reviewer for their detailed and constructive reviews! Based on their feedback, we have revised the manuscript and attached it to this response with the changes highlighted in blue. We have also included a detailed response to each reviewer's comments separately. Here is a summary of the changes made to the manuscript:

1. We have extended our problem formulation to consider time-varying connection probabilities.
2. We have removed our initial assumption that connection probabilities are known apriori and instead have used connection probability estimates that are updated over time.
3. We have included an additional baseline, n-Fleet DAgger, which implements network thresholding before Fleet-DAgger, and an additional network setting, Changing-Scarce, to test our allocation policies under changing network connectivities.
4. We have added theorems and definitions to provide theoretical bounds for our allocation policies.
5. We have included additional numerical results to understand why ASA and n-ASA policies outperform baselines.

Overall, these modifications significantly improve the formulation by incorporating time-variant connectivity, connection probability estimates, additional baselines and experiments, and additional analysis of results. While these updates are substantial, they were straightforward extensions of our general algorithmic framework, showcasing its robustness and adaptability.

---

### Decision · Program_Chairs · 2024-09-04

**Decision:**

Accept

**Comment:**

Strengths:
- Addressing a timely and challenging problem in the field.
- The paper is relatively well-written.
- The proposed idea is interesting and seems to be technically sound, somehow supported by evaluations.

Weaknesses:
- The extendability of problem fomulation needs to be better investigated.
- More detailed analysis of the results would be beneficial.
- The evaluation can be improved by comparing against more diverse types of baselines.
- The contribution over Fleet-Dagger needs to be more clearly described.